



# ORCHIMIC (v1.0), a microbe-driven model for soil organic matter decomposition designed for large-scale applications

Ye Huang[1], Bertrand Guenet[1], Philippe Ciais[1], Ivan A. Janssens[2], Jennifer L. Soong[2,3], Yilong Wang[1], Daniel Goll[1], Evgenia Blagodatskaya[4,5], Yuanyuan Huang[1]

[1]Laboratoire des Sciences du Climat et de l'Environnement, LSCE/IPSL, CEA-CNRS-UVSQ, Université Paris-Saclay, 91191 Gif-sur-Yvette, France
[2]University of Antwerp, Department of Biology, 2610 Wilrijk, Belgium
[3]Lawrence Berkeley National Laboratory, 94720, Berkeley, CA, USA
[4]Department of Agricultural Soil Science, University of Goettingen, Büsgenweg 2 37077 Göttingen, Germany
[5]Institute of Physicochemical and Biological Problems in Soil Science, 142290 Pushchino, Russia

*Correspondence to*: Ye Huang (ye.huang@lsce.ipsl.fr)

**Abstract.** The role of soil microorganisms in regulating soil organic matter (SOM) decomposition is of primary importance in the carbon cycle, and in particular in the context of global change. Modelling soil microbial community dynamics to simulate its impact on soil gaseous carbon (C) emissions and nitrogen (N) mineralization at large spatial scales is a recent research field with the potential to improve predictions of SOM responses to global climate change. We here present a SOM model called ORCHIMIC whose input data that are consistent with those of global vegetation models. The model simulates decomposition of SOM by explicitly accounting for enzyme production and distinguishing three different microbial functional groups: fresh organic matter (FOM) specialists, SOM specialists, and generalists, while implicitly also accounting for microbes that do not produce extracellular enzymes, *i.e.* cheaters. This ORCHIMIC model and two other organic matter decomposition models, CENTURY (based on first order kinetics and representative for the structure of most current global soil carbon models) and PRIM (with FOM accelerating the decomposition rate of SOM) were calibrated to reproduce the observed respiration fluxes from FOM and SOM and their possible interactions from incubation experiments of Blagodatskaya et al. (2014). Among the three models, ORCHIMIC was the only one that captured well both the temporal dynamics of the respiratory fluxes and the magnitude of the priming effect observed during the incubation experiment. ORCHIMIC also reproduced well the temporal dynamics of microbial biomass. We then applied different idealized changes to the model input data, *i.e.* a 5 K stepwise increase of temperature and/or a doubling of plant litter inputs. Under 5 K warming, ORCHIMIC predicted a 0.002 K$^{-1}$ decrease in the C use efficiency (defined as the ratio of C allocated to microbial growth to the sum of C allocated to growth and respiration) and a 3 % loss of SOC. Under the double litter input scenario, ORCHIMIC predicted a doubling of microbial biomass, while SOC stock increased by less than 1 % due to the priming effect. This limited increase in SOC stock contrasted with the proportional increase in SOC stock as modelled by the conventional SOC decomposition model (CENTURY), which cannot reproduce the priming effect. If temperature increased by 5 K and litter input is doubled, the model predicted almost the same loss of SOC as when only temperature was increased. These tests suggest that the responses of SOC stock to warming and increasing input may differ a lot from those simulated



by conventional SOC decomposition models, when microbial dynamics is included. The next step is to incorporate the ORCHIMIC model into a global vegetation model to perform simulations for representative sites and future scenarios.

## 1 Introduction

Soils contain the largest stock of organic carbon (C) in terrestrial ecosystems (MEA, 2005), ranging from 1220 to 2456 Pg C
(Batjes, 2014; Jobbagy and Jackson, 2000). Relatively small changes (< 1 %) in this global soil organic carbon (SOC) pool are therefore of similar order of magnitude as anthropogenic $CO_2$ emissions, and warming-induced SOC losses may thus represent a large feedback to climate change (Jenkinson et al., 1991). A realistic representation of SOC dynamics in Earth system models is therefore a necessity for accurate climate projections, and reducing the uncertainty of SOC stock responses to global climate change was put forward as a research priority (Arora et al., 2013; Friedlingstein, 2015).
In most Earth system models, the decomposition of soil organic matter (SOM) is represented by first order kinetics (Todd-Brown et al., 2013). The role of microbes during decomposition is not explicitly represented in these models, rather, the decomposition flux, modified by environmental factors, is dependent on the size of the substrate pool. However, these global models fail to accurately reproduce the observed global spatial distribution of SOC (Todd-Brown et al., 2013) even when adjusting parameters (Hararuk et al., 2014), suggesting structural problems in their formulations. One of the underlying
reasons might be that microbial community structure and activity are not explicitly represented (Creamer et al., 2015).
Typical SOC models distinguish rapidly decomposing from slowly decomposing plant litter and SOC pools. With first-order kinetics, the decomposition rate of each pool is independent from the other pools, as decomposition rates are decoupled from microbial dynamics. As a result, the priming effect, defined as changes in SOC decomposition rates induced by addition of fresh, energy-rich, organic matter (FOM) (Blagodatskaya and Kuzyakov, 2008), cannot be reproduced by these SOC models
(Guenet et al., 2016). However, priming effects have been widely observed in laboratory studies using different types of soil, with different types of FOM added during soil incubation experiments on timescales of less than one day to several hundred days (Fontaine et al., 2003; Kuzyakov and Bol, 2006; Tian et al., 2016), as well as in field experiments (Prévost-Bouré et al., 2010; Subke et al., 2004; Subke et al., 20011; Xiao et al., 2015). The influence of priming on SOC dynamics on long timescales, from years to decades, and at large spatial scales remains uncertain, but cannot be neglected in future SOC stock
simulations, considering the projected increase of plant litter inputs to soil in response to the fertilizing effects of elevated $CO_2$, globally increasing nitrogen (N) deposition, and lengthening growing seasons (Burk et al., 2017; Qian et al. 2010).
Soil microbial dynamics are believed to be responsible for the priming effect (Kuzyakov et al., 2000). Recently, new models have included the effects of microbial dynamics on SOC decomposition, but not always with an explicit representation of microbial processes (Schimel and Weintraub, 2003; Moorhead and Sinsabaugh, 2006; Lawrence et al., 2009; Wang et al.,
2013; Wieder et al., 2014; Kaiser et al., 2014, 2015; He et al., 2015). In those models, SOC decomposition is mediated by soil enzymes released by microorganisms (Allison et al, 2010; Schimel and Weintraub, 2003; Lawrence et al., 2009;). Although different groups of microorganisms can produce different enzymes, with large redundancy (Nannipieri et al 2003),





the production of enzymes in models is typically modelled as a fixed fraction of total microbial biomass (Allison et al., 2010; He et al., 2015) or as a fixed fraction of the uptake of C or N (Schimel and Weintraub, 2003; Kaiser et al., 2014, 2015). However, negative priming effects, *i.e.* reduced SOC decomposition in response to FOM addition, as occasionally observed in soil incubation experiments (Guenet et al., 2012; Hamer and Marschner, 2005; Tian et al., 2016), suggest that the

preferential production of enzymes decomposing FOM or an inhibited production of enzymes decomposing SOC is possible. Moreover, it has been reported that enzyme activity can be stimulated by substrate addition and be suppressed by nutrient addition (Allison and Vitousek, 2005). These observations suggest that the production of enzymes is modulated by substrate availability and quality, and not just by microbial uptake or microbial biomass.

Logically, SOC models ignoring microbial dynamics also do not distinguish between active and dormant microbial biomass,

thereby neglecting the fact that the different physiology of microbes during these two states (Wang et al., 2014). For instance, only active microbes are involved in decomposing SOC (Blagodatskaya and Kuzyakov, 2013) and in producing enzymes (He et al., 2015). However, with 80 % of microbial cells typically being dormant in soils, dormancy is the most common state of microbial communities (Lennon and Jones, 2011).  Reactivation of dormant microbes due to addition of labile substrates is one of the proposed mechanisms explaining the priming effect (Blagodatskaya and Kuzyakov, 2008).

Thus, explicitly representing the active fraction of microbial biomass, rather than the entire microbial biomass is a promising avenue to improve SOC models.

In previous models that explicitly simulate microbial dynamics, enzyme-mediated decomposition rates were modelled using Michaelis-Menten kinetics (Allison et al., 2010), or reverse Michaelis-Menten kinetics (Schimel and Weintraub, 2003; Lawrence et al., 2009). Michaelis-Menten kinetics were also used to model uptake of C by microbes (Allison et al., 2010). In

comparison to these two formulations, first-order accurate equilibrium chemistry approximation (ECA) kinetics, performed better than Michaelis-Menten kinetics for a single microbe feeding on multiple substrates or for multiple microbes competing for multiple substrates (Tang and Riley, 2013). The ECA kinetics combine the advantages of Michaelis-Menten and reverse Michaelis-Menten kinetics (Tang, 2015), making this formulation more suitable for application in conceptual microbial models.

Nutrient dynamics are often ignored in SOC models (Allison et al., 2010; Wang et al., 2013; Wang et al., 2014; He et al., 2015; Guenet et al., 2016), in particular in the SOC models used with Earth system models (Anav et al., 2013), despite the fact that nutrients can be a rate-limiting for many biological processes in ecosystems (Vitousek and Howarth, 1991). By providing rhizosphere microbes with energy-rich, nutrient-poor exudates, roots may elicit microbial growth, their need for nutrients, and subsequently their production of SOC-decomposing enzymes. Nutrient availability, especially that of the

macro-element nitrogen (N), thus regulates the priming effect of microbes in response to root exudation (Janssens et al., 2010).  Including N dynamics in SOC models is therefore also a necessity for accurate projections of future SOC stocks.

In this study, a microbe-driven SOM decomposition model -ORCHIMIC- is described and tested against incubation experiment results. In ORCHIMIC, enzyme production is dynamic and depends on the availability of carbon and nitrogen in FOM and SOM substrates and on a specific pool of available C and N. Three microbial function types (MFTs) - generalists,



FOM specialists and SOM specialists- are included, along with explicit representation of their dormancy, while a fraction of these microbes being cheaters do not invest in producing SOC decomposing enzymes themselves, but profit from the investments of others. ORCHIMIC is developed with the aim of being incorporated in a global land surface model. In thus study it was embedded in the ORCHIDEE land biosphere model (Krinner et al., 2005), but any other global land surface

model could also be used for grid-based simulations.

The ORCHIMIC is described in Sect. 2 and the two conceptually simpler models -a first order kinetics model called CENTURY, which was derived from Parton et al., (1987) and constitutes the SOC decomposition module of the ORCHIDEE model (Krinner et al., 2005), and a first order kinetics model called PRIM, which is a variant of the CENTURY model modified to include interactions between pools to enable representation of priming of decomposition rates (Guenet et

al., 2016)- are described in Sect. 3. The model parameters were calibrated against soil incubation data from Blagodatskaya et al. (2014) (Sect. 4). Different idealized tests of the ORCHIMIC model response including doubling FOM input and/or a stepwise increase of temperature were performed (Sect. 5).

## 2 ORCHIMIC description

The ORCHIMIC model is zero-dimensional and considers biology and soil physics homogenous within the soil grid to

which it is applied. The model simulates C and N dynamics at a daily time step. Inputs of the model are additions of C and N from plant litter or from other sources, and plant uptake of N. In return, the model predicts soil carbon and nitrogen pools and respired $CO_2$ fluxes.

A total of 11 pools are considered for both C and N (Fig. 1). The two FOM pools are metabolic ($LM$) and structural ($LS$) plant litter. The three SOM pools are the active ($SA$), slow ($SS$) and passive ($SP$) pools with short, medium and long turnover

time (Parton et al., 1987). $SA$ consists of dead microbes and deactivated enzymes with short turnover time. $SS$ contains SOM generated during decomposition of litter and $SA$ pool, which are chemically more recalcitrant and/or physically protected with medium turnover time. $SP$ is a pool of SOM generated during decomposition of SOM in other pools which are most resistant to decomposition with long turnover time. The outgoing C and N fluxes from the substrate pools include the decomposition of the FOM pools by $EF$ enzymes and of the SOM pools by $ES$ enzymes, as well as fluxes between FOM and

SOM pools that represent physicochemical protection mechanisms, such as occlusion of substrates in macro-aggregates.

The available pools ($Avail$) represent C and N that are directly available to microbes. The $Avail$ pool receives inputs from substrate decomposition, desorption from mineral surfaces, microbial mortality and decay. The $Avail$ pool is depleted by uptake of C and N by active microbes, adsorption on mineral surfaces, and leaching losses. The $Adsorb$ pool represents C and N that are unavailable to microbes because of adsorption by mineral surfaces.

Four MFTs, including SOM specialists, FOM specialists, generalists and cheaters, are explicitly or inexplicitly represented, as described in Sect. 2.1. Each MFT is further divided in active ($BA$) and dormant ($BD$) biomass. The outgoing C fluxes from active microbes are growth respiration, maintenance respiration, overflow respiration, dormancy, death and enzyme





production. During dormancy, death and enzyme production, corresponding amount of N are also lost from active microbes. N is also released from active microbes when maintenance respiration is at cost of their own biomass. Dormant microbes can be reactivated (a flux of C and N from dormant to active microbes) and lose C and release N during maintenance respiration but at a slower rate compared to active microbes.

The two enzyme pools include enzymes that can decompose either FOM ($EF$) or SOM ($ES$). Enzyme pools receive inputs through microbial enzyme production, and decline through enzyme turnover. The equations corresponding to each process shown in Fig. 1 are given in Sect. 2.2, and for fluxes between pools in Sect. 2.3.

### 2.1 Microbial functional types

Four MFTs, SOM specialists, FOM specialists, generalists and cheaters, are represented with a set of parameters, including a
MFT specific C/N ratio ($BCN_i$) and maximum uptake rate of C ($Vmax_{uptak,i}$) for the $i^{th}$ MFT, optimum soil moisture ($\theta_0$) and $pH$ ($pH_0$) for microbial uptake, parameters controlling the microbial uptake sensitivities to soil moisture ($\theta_s$) and $pH$ ($pH_s$), maximum enzyme production coefficient ($K_e$), the ability to produce FOM specific enzymes $EF$ ($EFr_i$) and SOM specific enzymes $ES$ ($ESr_i$) and the dissolvable fraction of dead microbial biomass ($s_C$ for C and $s_N$ for N) (Table 1). Generalists, SOM specialists and FOM specialists are the three enzyme-producing MFTs that are explicitly considered. The main
differences among them are their C/N ratio and their maximum capacity to produce enzymes for decomposing specific pools. The C/N ratios $BCN_i$ are set to 4.59 and 8.30 for SOM and FOM specialists, respectively (Mouginot et al., 2014), based on the assumption that SOM decomposers are mainly bacteria and FOM decomposers are mainly fungi (Kaiser et al., 2014). The C/N ratio of generalists is set to 6.12, in-between that of FOM and SOM specialists. The maximum total enzyme production capacities are set to be the same for each MFT. FOM specialists ($i$=1) can produce more enzymes that decompose
FOM ($EFr_1:ESr_1$=0.75:0.25). SOM specialists ($i$=2) can potentially produce more enzymes that decompose SOM ($EFr_2:ESr_2$=0.25:0.75), and generalists ($i$=3) can potentially produce both kinds of enzymes in equal proportions ($EFr_3:ESr_3$=0.5:0.5). However, the real production of the two enzymes depends on availability of substrates and available C. Cheaters are microbes that do not produce substrate-decomposing enzymes but profit from the enzymes produced by the other MFTs (Allison, 2005; Kaiser et al., 2015). Cheaters were added as an explicit microbial functional group in an
individual-based micro-scale microbial community model with an explicit positioning of microbes to access substrate (Kaiser el al., 2015). However, this approach is only feasible in a micro scale model, because the coexistence of cheaters and enzyme-producing microbes in situ is only sustainable in heterogeneous environments. In non-spatially explicit zero-dimensional models, like ORCHIMIC that assumes a homogeneous environment, cheaters will always have a competitive advantage over other microbes in taking up C and N, because they do not have to invest in enzyme production. This will
eventually drive enzyme producing MFTs to extinction at steady state (Allison, 2005), and the model would crash because enzymes are no longer produced and all microbes eventually die. In ORCHIMIC, enzyme production per unit of active microbial biomass decreases with increasing available C availability (see Sect. 2.3.7 for this dynamic enzyme production mechanism). This corresponds to a larger fraction of microbial biomass behaving as cheaters than when considering that





enzyme production per unit of non-cheaters is constant. Because all three MFTs that are explicitly represented can partly act as cheaters, and do so to variable degree, cheaters are a fourth MFT that is inexplicitly included in the model.

### 2.2 Carbon and nitrogen pools

#### 2.2.1 Litter pools

The two FOM pools, *LM* and *LS*, receive prescribed inputs from plant litter fall. The distribution of FOM carbon between the *LM* and *LS* compartments is a prescribed function of the lignin to N ratio of plant material (Eq. (1)) after Parton et al., (1987) (see Sect. 2.3.1). The C/N ratio of the *LS* pool is set to 150 (Parton et al., 1988) and the C/N ratio of the *LM* pool is variable depending on the C/N ratio of the FOM input (a forcing of ORCHIMIC representing litter quality). The dynamics of the FOM pools are described by:

$$LMf = 0.85 - 0.018 \times \frac{LLf_{in}}{LCN_{in}} \tag{1}$$

$$\frac{dLM_X}{dt} = LM_{X,in} - D_{X,LM} \tag{2}$$

$$\frac{dLS_X}{dt} = LS_{X,in} - D_{X,LS} \tag{3}$$

where *LMf* is the fraction of litter input C allocated to the *LM* pool; $LLf_{in}$ and $LCN_{in}$ are the lignin content and C/N ratio of litter input to the FOM pools; X represents C or N; $LM_{x,in}$ and $LS_{x,in}$ are the litter input partitioned to the *LM* and *LS* pools

based on *LMf* and C/N ratio of litter input, respectively; $D_{X,LM}$ and $D_{X,LS}$ are loss of X due to the enzymatic decomposition of *LM* and *LS*, respectively (see Sect. 2.3.1).

#### 2.2.2 Soil organic matter pools

The three SOM pools (*SA*, *SS* and *SP*) represent substrates that are decomposed by SOM decomposing enzymes. The *SA* represents the insoluble part of dead microbes and deactivated enzymes that have a fast turnover time. The dynamics of this

pool are described by:

$$\frac{dSA_X}{dt} = \sum_i \left[ BAd_{X,i} \times (1 - s_X) \right] + \sum_i \left( EFd_{X,i} + ESd_{X,i} \right) - D_{X,SA} \tag{4}$$

where the first term on the right of the equation represents input from non-soluble active microbial biomass mortality summed over all the MFTs, $BAd_{X,i}$ is the input of C or N due to mortality of MFT *i* (see Sect. 2.3.6); $s_X$ is the proportion of microbial biomass that is soluble; the second term represents the input from enzymes that lost their activity; $EFd_{X,i}$ and $ESd_{X,i}$

are the inputs of C or N due to turnover of *EF* and *ES* enzymes, respectively, produced by MFT *i* (see Sect. 2.3.7); $D_{X,SA}$ is the loss of C or N due to decomposition of *SA* (see Sect. 2.3.1).

For *SS* pool, there is a flux going from the FOM pool to the *SS* pool without being processed by the microbes. Following the CENTURY model (Parton et al., 1987; Stott et al., 1983), 70 % of lignin in *LS* is assumed to go to the *SS* pool without





microbial uptake. *LtoSS* is the fraction of decomposed *LM* and non-lignin *LS* that goes into the *SS* pool. Similarly, there is also a flux coming from the *SA* to the *SS* pool that represents non-biological SOM protection processes, such as physical protection (Von Lutzow et al., 2008). The dynamics of the *SS* pool are given by:

$$\frac{dSS_X}{dt} = D_{X,LM} \times LtoSS + D_{X,LS} \times (1 - LLf) \times LtoSS + D_{X,LS} \times LLf \times 0.7 + D_{SX,A} \times SAtoSS - D_{X,SS} \qquad (5)$$

where the first term represents input of X (C or N) from the *LM* pool without microbial processing; the second and third terms represent input from the non-lignin part and the lignin part of the *LS* pool, respectively; the fourth term represents input from the *SA* pool; *LLf* is the lignin fraction of the *LS* pool; $D_{X,SS}$ is the loss of C or N from the decomposition of the *SS* pool (see Sect. 2.3.1) and *SAtoSS* is the fraction of decomposed *SA* becoming physically or chemically protected and added to the *SS* pool, as modified by the soil clay content (*CC*) (Parton et al., 1987):

$$SAtoSS = 0.146 + 0.68 \times C \qquad (6)$$

The *SP* pool is more resistant to decomposition than *SS*. It receives fluxes from the *SA* and *SS* pools and its dynamics are described by:

$$\frac{dSP_X}{dt} = D_{X,SA} \times SAtoSP + D_{X,SS} \times SStoSP - D_{X,SP} \qquad (7)$$

where the first and second terms represent input from the *SA* and *SS* pools, respectively; *SAtoSP*=0.004 and *SStoSP*=0.03 are

the fractions of decomposed *SA* and *SS* that go into the *SP* pool, respectively (Parton et al., 1987); $D_{X,SP}$ is the loss of C or N due to decomposition (see Sect. 2.3.1).

**2.2.3 Pools of C and N available for microbial and plant uptake, and gaseous N loss**

The available C and N pool (*Avail* in Fig. 1) represents C and N directly available for microbial uptake. It receives C and N decomposed from FOM and SOM pools, and the soluble part of dead microbes (Schimel & Weintraub, 2003; Kaiser et al.,

2014) and C and N desorbed from mineral surfaces. C and N from this pool can be taken up by microbes or adsorbed onto mineral surfaces. N released from microbial biomass after maintenance respiration of dormant microbes and active microbes, only when C uptake is not enough, is also assumed to be an input to the *Avail* pool. In addition, uptake of N by plant roots (a forcing of ORCHIMIC in the case of coupling with a vegetation model) and loss of C and N due to leaching are modelled as fluxes removed from this pool. Gaseous N loss due to nitrification and denitrification (see Eq. (26) in Sect. 2.3.1) is

considered as a decreased input from substrate decomposition. The dynamics of the *Avail* pool are described by Eq. (8) and (9) for C and N, respectively.

$$\frac{dAvail_C}{dt} = D_{C,LM} \times (1 - LtoSS) + D_{C,LS} \times LLf \times 0.3 + D_{C,LS} \times (1 - LLf) \times (1 - LtoSS) + D_{C,SA} \times (1 - SAtoSS - SAtoSP) +$$
$$D_{C,SS} \times (1 - SStoSP) + D_{C,SP} + \sum_i (BAd_{C,i} \times s_C) - \sum_i Uptakeadj_{C,i} + Desorb_{Adsorb,C} - Adsorb_{Avail,C} -$$
$$leaching_C \qquad (8)$$




$$\frac{dAvail_N}{dt} = D_{N,LM} \times (1 - LtoSS) + D_{N,LS} \times LLf \times 0.3 + D_{N,LS} \times (1 - LLf) \times (1 - LtoSS) + D_{N,SA} \times (1 - SAtoSS - SAtoSP) +$$

$$D_{N,SS} \times (1 - SStoSP) + \left(D_{N,SP} - \sum_j Dloss_{N,j}\right) + \sum_i (BAd_{N,i} \times s_N) - \sum_i BAg_{N,i} + Desorb_{Adsorb,N} -$$

$$Desorb_{Avail,N} + \sum_i (BAm_{N,i} + BDm_{N,i}) - Veg_{uptake,N} - leaching_N \tag{9}$$

where $Uptakeadj_{C,i}$ is C taken up by MFT $i$ (see Sect. 2.3.2); $Desorb_{Adsorb,C}$ and $Desorb_{Adsorb,N}$ is the flux of C and N desorbed

from mineral surface, respectively (see Sect. 2.3.8); $Adsorb_{Avail,C}$ and $Adsorb_{Avail,N}$ are C and N absorbed by mineral surface, respectively (see Sect. 2.3.8); $Dloss_{N,j}$ is the gaseous N loss; $BAg_{N,i}$ is N assimilated by MFT $i$ (see Sect. 2.3.4); $BAm_{N,i}$ and $BDm_{N,i}$ is N released from maintenance respiration of active and dormant biomass for MFT $i$ to $Avail_N$ pool, respectively (see Sect. 2.3.3); $Veg_{uptake,N}$ is N uptaken by plants, a boundary condition of the model; $leaching_C$ and $leaching_N$ are the loss of C and N due to leaching, respectively.

### 2.2.4 Adsorbed C and N on mineral surfaces

The C and N in the *Avail* pool can be reversibly adsorbed (*Adsorb* pool in Fig. 1) and rendered unavailable to microbes and plants (for N). The dynamics of the *Adsorb* pool are given by:

$$\frac{dAdsorb_X}{dt} = Adsorb_{Avail,X} - Desorb_{Adsorb,X} \tag{10}$$

where the first term is the C or N adsorbed onto mineral surface and the second term is the C or N desorbed from mineral

surface (see Sect. 2.3.8).

### 2.2.5 Enzymes pools

We distinguish between two types of enzymes (*EF* and *ES*), which catalyse the decomposition of FOM and SOM, respectively. Each MFT produces enzymes according to their specialization. The turnover rate of both types of enzymes is assumed to be the same. The dynamics of the FOM and SOM decomposing enzyme pools are described by:

$$\frac{dEF_{X,i}}{dt} = EFg_{X,i} - EFd_{X,i} \tag{11}$$

$$\frac{dES_{X,i}}{dt} = ESg_{X,i} - ESd_{X,i} \tag{12}$$

where $EFg_{X,i}$ and $ESg_{X,i}$ are the production rates of enzymes *EF* and *ES* by MFT $i$, with $i = 1$ for FOM specialists, $i = 2$ for FOM specialists and $i = 3$ for generalists, respectively (see Sect. 2.3.7). $EFd_{X,i}$ and $ESd_{X,i}$ are the turnover rates of the enzymes *EF* and *ES*, produced by MFT $i$ (see Sect. 2.3.7).

### 2.2.6 Active and dormant microbial biomass pools

In ORCHIMIC, each MFT can be active or dormant and can switch from one state to the other depending on environmental conditions. When active, the mass of each MFT is defined by the balance between their growth, death, production of



enzymes, maintenance and growth respiration and exchange of mass with dormant biomass ($BD$). If the uptake of C cannot meet the need for maintenance respiration, the active mass of a MFT will respire part of its biomass as $CO_2$. When microbial biomass becomes dormant, its carbon can be reactivated or respired through maintenance respiration. When respiration is at the cost of their biomass, a corresponding amount of N is assumed to be lost from dormant microbial biomass and goes to

the *Avail* pool so that the stoichiometry of the dormant microbes remains unchanged. The dynamics for active and dormant microbial MFTs are described by:

$$\frac{dBA_{X,i}}{dt} = BAg_{X,i} + B_{DtoA,X,i} - BAd_{X,i} - EFg_{X,i} - ESg_{X,i} - B_{AtoD,X,i} - BAm_{X,i} \tag{13}$$

$$\frac{dBD_{X,i}}{dt} = B_{AtoD,X,i} - BDm_{X,i} - B_{DtoA,X,i} \tag{14}$$

where $BAg_{X,i}$ is the increase of $BA_X$ due to growth for MFT $i$ (see Sect. 2.3.4); $B_{DtoA,X,i}$ is the X in microbes transformed from

dormant state to active state for MFT $i$; $B_{AtoD,X,i}$ is the X in microbes transformed from active state to dormant state for MFT $i$ (see Sect. 2.3.5); $BAd_{X,i}$ is the loss of X due to death of active biomass of MFT $i$; $BAm_{X,i}$ and $BDm_{X,i}$ are the loss of X in active biomass and dormant biomass, respectively, due to maintenance respiration of MFT $i$.

### 2.3 Modelling the processes controlling fluxes between pools

### 2.3.1 Organic matter decomposition

The substrate used by microorganisms includes FOM and SOM. The FOM and SOM pools are decomposed by enzymes *EF* and enzymes *ES*, respectively. The decomposition process is modelled using a combination of Arrhenius and Michaelis-Menten equations (Allison et al., 2010), with different *Vmax* values for each substrate pool and different Michaelis-Menten constants (*KM*) for FOM and SOM. To avoid unrealistic decomposition rates when enzyme concentrations are high, an enzyme-dependent term was added in the denominator (ECA kinetics). *Vmax* values are considered to be sensitive to

temperature and modelled using an Arrhenius equation (Eq. (16)), with higher activation energy (*Ea*) for more recalcitrant substrates (Allison et al., 2010). *KM* is also considered to be sensitive to temperature (Allison et al., 2010; Wang et al., 2013) and the dependency of *KM* on temperature is modelled using an Arrhenius equation with activation energy ($Ea_{KM}$) of 30 kJ mol$^{-1}$ (Davidson and Janssens., 2006). All decomposition functions are modulated by soil moisture ($\theta$) and *pH*. The decomposition function of *LS* is further modified by its lignin content (Parton et al., 1987). The decomposition function of

*SA* is further modified by soil clay content (*CC*) (Parton et al., 1987). The functions modifying substrates' decomposition rates by $\theta$, *T*, *pH*, lignin content and soil clay content are given by:

$$F_\theta = max[0.25, min\,(1, -1.1 \times \theta^2 + 2.4 \times \theta - 0.29)] \tag{15}$$

$$F_{T,j} = e^{\frac{-Ea_j}{R}\left(\frac{1}{T} - \frac{1}{T_{ref}}\right)} \tag{16}$$





$$F_{pH} = e^{\frac{-(pH-pH_{0,ENZ})^2}{pH_{s,ENZ}{}^2}} \tag{17}$$

$$F_{lignin} = e^{-3 \times LLf} \tag{18}$$

$$F_{clay} = 1 - 0.75 \times CC \tag{19}$$

where $F_\theta$, $F_{T,j}$, $F_{pH}$, $F_{clay}$ and $F_{lignin}$ are functions of soil moisture ($\theta$), temperature ($T$), pH, clay content ($CC$), lignin content
5 ($LLf$) that modify substrate decomposition rates; $j$ represents substrate which are $LM$, $LS$, $SA$, $SS$ or $SP$; $Ea_j$ is activation
energy of substrate $j$; $Tref$ is a reference temperature which was set to 285.15 K; $pH_{0,ENZ}$ is the optimum $pH$ of enzymatic
decomposition; $pH_{s,ENZ}$ is a sensitivity parameter of enzymatic decomposition; $R$ is the ideal gas constant (0.008314 kJ mol[-1]
K[-1]).

Thus, the decomposition of C in $LM$, $LS$, $SA$, $SS$ and $SP$ pools can be described by Eq. (20), (21), (22), (23) and (24),
10 respectively. The decomposition of N follows the C/N ratio of the corresponding substrate (Eq. (25)). N can be lost through
volatilization of N products (NH$_3$, N$_2$, N$_2$O) generated during decomposition, nitrification and denitrification (Schimel, 1986;
Mosier et al., 1983). Like in CENTURY model (Parton et al., 1987, 1988), we assumed that 5 % of total N mineralized
during decomposition is lost to the atmosphere as a first order approximation of volatilization, nitrification, and
denitrification losses ($Dloss_{N,j}$, Eq. (26)).

$$D_{C,LM} = Vmax_{LM} \times F_{T,LM} \times \sum_i EF_i \times \frac{LM_C}{KM_F \times e^{\frac{-Ea_{KM}}{R} \times \left(\frac{1}{T} - \frac{1}{T_{ref}}\right)} + LM_C + \sum_i EF_{C,i}} \times F_\theta \times F_{pH} \times dt \tag{20}$$

$$D_{C,LS} = \frac{Vmax_{LM}}{Adj_{LS}} \times F_{T,LS} \times \sum_i EF_{C,i} \times \frac{LS_C}{KM_F \times e^{\frac{-Ea_{KM}}{R} \times \left(\frac{1}{T} - \frac{1}{T_{ref}}\right)} + LS_C + \sum_i EF_{C,i}} \times F_\theta \times F_{pH} \times F_{lignin} \times dt \tag{21}$$

$$D_{C,SA} = Vmax_{SS} \times Adj_{SA} \times F_{T,SA} \times \sum_i ES_{C,i} \times \frac{SA_C}{KM_S \times e^{\frac{-Ea_{KM}}{R} \times \left(\frac{1}{T} - \frac{1}{T_{ref}}\right)} + SA_C + \sum_i ES_{C,i}} \times F_\theta \times F_{pH} \times F_{clay} \times dt \tag{22}$$

$$D_{C,SS} = Vmax_{SS} \times F_{T,SS} \times \sum_i ES_{C,i} \times \frac{SS_C}{KM_S \times e^{\frac{-Ea_{KM}}{R} \times \left(\frac{1}{T} - \frac{1}{T_{ref}}\right)} + SS_C + \sum_i ES_{C,i}} \times F_\theta \times F_{pH} \times dt \tag{23}$$

$$D_{C,SP} = \frac{Vmax_{SS}}{Adj_{SP}} \times F_{T,SP} \times \sum_i ES_{C,i} \times \frac{SP_C}{KM_S \times e^{\frac{-Ea_{KM}}{R} \times \left(\frac{1}{T} - \frac{1}{T_{ref}}\right)} + SP_C + \sum_i ES_{C,i}} \times F_\theta \times F_{pH} \times dt \tag{24}$$

$$D_{N,j} = D_{C,j} \times \frac{j_N}{j_C} \tag{25}$$

$$Dloss_{N,j} = D_{N,j} \times 0.05 \tag{26}$$





where $D_{C,LM}$, $D_{C,LS}$, $D_{C,SA}$, $D_{C,SS}$ and $D_{C,SP}$ are C flux from $LM$, $LS$, $SA$, $SS$, and $SP$ pools due to enzymatic decomposition, respectively; $D_{N,j}$ is the N flux from substrate $j$ due to enzymatic decomposition; $Dloss_{N,j}$ is the gaseous N loss from substrate $j$; $j$ represents substrate which are $LM$, $LS$, $SA$, $SS$ or $SP$; $Vmax_{LM}$ and $Vmax_{SS}$ are maximum decomposition rates of C in $LM$ and $SS$ pool, respectively; $KM_F$ and $KM_S$ are $KM$ for FOM and SOM pools, respectively; $dt$ is the time step in unit of hour;

$Adj_{LS}$ is the ratio of maximum decomposition rate of C in $LM$ to that in $LS$; $AdjSA$ and $AdjSP$ is the ratio of maximum decomposition rate of C in $SA$ to that in $SS$ and that in $SS$ to that in $SP$, respectively; $j_C$ and $j_N$ are the mass concentrations of C and N in substrate $j$ pool, respectively.

### 2.3.2 Uptake of C and N by microbes

The uptake of C from the $Avail$ pool is modelled as a function of microbial active biomass (Wang et al., 2014), and uptake

rates are modulated by $T$, $\theta$ and $pH$. The effect of $T$ on the uptake rate is modelled using an Arrhenius equation following Allison et al. (2010). The effect of $\theta$ and $pH$ are modelled using exponential-quadratic functions (Reth et al., 2005). Additionally, the uptake rate is also affected by the saturation ratio of the available C pool ($Avail_C$) they feed on. ECA kinetics formulation (Tang and Riley, 2013) is used to estimate the saturation ratio of the $Avail$ pool. With this formula, the saturation ratio depends not only on the concentration of the $Avail$ pool but also on the concentration of the active microbial

biomass. Thus, competition for the $Avail$ pool among different MFTs and limitation for one MFT is implicitly included because the uptake rate is modulated by active biomass concentration and the level of the $Avail$ pool. Therefore, when active biomass is high, the uptake rate per unit of active biomass is reduced, mimicking the competition. The functions modifying microbes' uptake rates by $T$, $\theta$ and $pH$ are given by Eq. (27), (28) and (29), respectively. The saturation ratio of the available C pool is given by Eq. 30.

$$f_{T,i} = e^{\frac{-Ea_{uptake}}{R}\left(\frac{1}{T} - \frac{1}{T_{ref}}\right)} \tag{27}$$

$$f_{\theta,i} = e^{\frac{-\left(\theta - \theta_{0,i}\right)^2}{\theta_{s,i}^2}} \tag{28}$$

$$f_{pH,i} = e^{\frac{-\left(pH - pH_{0,i}\right)^2}{pH_{s,i}^2}} \tag{29}$$

$$\Phi_{C,i} = \frac{Avail_C}{KM_{uptake,C,i} \times e^{\frac{-Ea_{KM}}{R} \times \left(\frac{1}{T} - \frac{1}{T_{ref}}\right)} + Avail_C + \sum_i BA_{C,i}} \tag{30}$$

where $f_{T,i}$, $f_{\theta,i}$ and $f_{pH,i}$ are temperature, soil moisture and $pH$ function modifying uptake rate of MFT $i$; $\Phi_{C,i}$ is the saturation

ratio of the available carbon pool; $Ea_{uptake}$ is the activation energy for uptake; $\theta_{0,i}$ and $pH_{0,i}$ are optimum soil moisture and $pH$ for uptake by MFT $i$, respectively; $\theta_{s,i}$ and $pH_{s,i}$ are sensitivity parameters for uptake by MFT $i$ to soil moisture and $pH$, respectively.



Potential uptake of C is given by Eq. (31). Total uptake of C by all microbes should not exceed the total available C, therefor all microbes decrease their uptake at the same proportion as a trade off when total demand of C is larger than the total available C (Eq. (32)).

$$Uptake_{C,i} = Vmax_{uptake,C,i} \times \Phi_{C,i} \times BA_{C,i} \times f_{T,i} \times f_{\theta,i} \times f_{pH,i} \times dt \tag{31}$$

$$Uptakeadj_{C,i} = \begin{cases} Uptake_{C,i}, \sum_i Uptake_{C,i} \leq TAvail_C \\ Uptake_{C,i} \times \frac{TAvail_C}{\sum_i Uptake_{C,i}}, \sum_i Uptake_{C,i} > TAvail_C \end{cases} \tag{32}$$

The total available C or N includes the C and N in the *Avail* pool as well as those rendered available during decomposition and those recycled from deceased microbes (Eq. (33) and Eq. (34)).

$$TAvail_C = Avail_C + \sum_i (BAd_{C,i} \times S_C) + \sum_j D_{C,j} \tag{33}$$

$$TAvail_N = Avail_N + \sum_i (BAd_{N,i} \times S_N) + \sum_j (D_{N,j} - Dloss_{N,j}) \tag{34}$$

The uptake of N by microbes follows the C/N ratio of total available C and N (Eq. 35).

$$Uptakeadj_{N,i} = Uptakeadj_{C,i} \times \frac{TAvail_N}{TAvail_C} \tag{35}$$

where $Uptake_{C,i}$ is the theoretical uptake of C by MFT *i* under given $\Phi_{C,i}$ without considering the total available C; $Uptakeadj_{C,i}$ and $Uptakeadj_{N,i}$ are the real uptake of C and N by MFT *i*, respectively; the $KM_{uptake,C,i}$ is *KM* for uptake of C by MFT *i* and it is set to be the same for all MFTs; $Vmax_{uptake,C,i}$ is the maximum uptake rate of C by MFT *i* and it is also set to
be the same for all MFTs; $TAvail_C$ and $TAvail_N$ are the total available C and N, respectively.

### 2.3.3 Maintenance respiration

The maintenance respiration of MFTs (bacteria and fungi) is modelled as a fixed ratio (maintenance respiration coefficient) of their biomass (Schimel & Weintraub, 2003; Lawrence et al., 2009; Allison et al., 2010; Wang et al., 2014; He et al., 2015) modulated by temperature using an Arrhenius equation following Tang and Riley (2015) (Eq. 36). Dormant microbes still
need a minimum of energy for maintenance, albeit at a much lower rate compared that of active microbes (Lennon and Jones, 2011). The maintenance respiration coefficient of dormant microbes is set to be a ratio *b* (between 0 and 1) of that of active microbes (Wang et al., 2014; He et al., 2015). The maintenance respiration thus can be described by Eq. (37) and (38) for active and dormant microbes, respectively. Dormant microbes respire their own biomass for survival (Eq. (39) and (40)). Active microbes take up C from the *Avail* C pool to meet their maintenance respiration requirement. If the C taken up does
not suffice, active microbes will use part of their own biomass for maintenance respiration (Eq. (41) and (42)).

$$Kr = Kr_{ref} \times e^{\frac{-Ea_{main}}{R} \left( \frac{1}{T} - \frac{1}{T_{ref}} \right)} \tag{36}$$





$$RAm_i = Kr \times BA_{C,i} \times dt \tag{37}$$

$$RDm_i = b \times Kr \times BD_{C,i} \times dt \tag{38}$$

$$BDm_{C,i} = RDm_i \tag{39}$$

$$BDm_{N,i} = \frac{BDm_{C,i}}{BCN_i} \tag{40}$$

$$BAm_{C,i} = \begin{cases} 0 & , RAm_i \leq Uptake_{adj_i} \\ RAm_i - Uptakeadj_{C,i}, & RAm_i > Uptakeadj_{C,i} \end{cases} \tag{41}$$

$$BAm_{N,i} = \frac{BAm_{C,i}}{BCN_i} \tag{42}$$

where $RAm_i$ and $RDm_i$ are maintenance respiration of active and dormant biomass for MFT $i$, respectively; $BDm_{C,i}$ and $BDm_{N,i}$ are C and N loss from dormant biomass for MFT $i$ due to maintenance respiration, respectively; $BAm_{C,i}$ and $BAm_{N,i}$ are C and N loss from active biomass for MFT $i$ due to maintenance respiration, respectively; $Ea_{main}$ is the activation energy of maintenance respiration coefficient; $Kr_{ref}$ and $Kr$ are the maintenance respiration coefficient at temperature $T$ and $T_{ref}$, respectively.

### 2.3.4 Growth of microbes, growth respiration and overflow respiration

If the C uptake exceeds the maintenance respiration flux, the excess of C can be allocated to microbial growth and growth respiration. The allocation between biomass production and growth respiration is controlled by the carbon assimilation efficiency (CAE), defined as the maximum fraction of C taken up that can be allocated to microbial biomass. The allocation of N uptake to microbial biomass is controlled by the nitrogen assimilation efficiency (NAE), which is defined as maximum fraction of N uptake that can be allocated to microbial biomass and is assumed equal to 1 (Manzoni and Porporato, 2009; Porporato et al., 2003). The final growth of microbial biomass depends on the availability of C and N and is limited by C or N depending on which element is more limiting. Growth of microbial biomass and growth respiration are described by Eq. (43)-(46) and (47), respectively. Under C limited conditions, the excess N in the microbes is released back to the *Avail* pool. Under N limited conditions, the C that cannot be incorporated in microbes is assumed to be respired through overflow metabolism (Eq. (48), *Schimel and Weintraub, 2003*), defined as overflow respiration.

$$g_{C,i} = \begin{cases} (Uptakeadj_{C,i} - RAm_i) \times CAE, & if\ Uptakeadj_{C,i} - RAm_i > 0 \\ 0 & , if\ Uptakeadj_{C,i} - RAm_i \leq 0 \end{cases} \tag{43}$$

$$g_{N,i} = \frac{Uptakeadj_{N,i} \times NAE}{BCN_i} \tag{44}$$

$$BAg_{C,i} = minimum(g_{C,i}, g_{N,i}) \tag{45}$$





$$BAg_{N,i} = \frac{BAg_{C,i}}{BCN_i} \qquad (46)$$

$$Rg_i = BAg_{C,i} \times \frac{1-CAE}{CAE} \qquad (47)$$

$$Ro_i = Uptakeadj_{C,i} - RAm_i - BAg_{C,i} - Rg_i \qquad (48)$$

where $g_{C,i}$ and $g_{N,i}$ are theoretical growth rates when only considering C-limited and N-limited growth rates, respectively;
$BAg_{C,i}$ and $BAg_{N,i}$ are the increase of C and N in microbial biomass, respectively; CAE is carbon assimilation efficiency;
NAE is nitrogen assimilation efficiency, which is set to 1 in this study; $Rg_i$ is growth respiration by MFT $i$; $Ro_i$ is overflow
respiration by MFT $i$.

**2.3.5 Transformation between active and dormant states**

Microbes can be active and dormant in the environment and can transform between these two states (Blagodatskaya and
Kuzyakov, 2013). Active microbes take up carbon and invest it in maintenance, growth, and enzyme production. Microbes
become dormant to lower their maintenance cost and survive under unfavourable conditions. The maintenance energy cost is
thought to be one of the key factors regulating the dormancy strategy (Lennon & Jones, 2011). Wang et al. (2014) assumed
that transformation between the two states was determined by the saturation ratio of substrates and the maintenance rate of
active microbes. In ORCHIMIC, microbes feed on the *Avail* pool instead of on substrates as in their model and considering
that C is the sole energy source, the saturation ratio of substrate is replaced here by the saturation ratio of the *Avail*$_C$ pool
($\Phi_{C,i}$). With $\Phi_{C,i}$, the effect of competition on the microbes' dormancy strategy is implicitly included. The transformation
from active to dormant phase ($B_{AtoD,X,i}$) or the reverse ($B_{DtoA,X,i}$) are given by:

$$B_{AtoD,X,i} = \Phi_{C,i} \times Kr \times BA_{X,i} \times dt \qquad (49)$$

$$B_{DtoA,X,i} = (1 - \Phi_{c,i}) \times Kr \times BD_{X,i} \times dt \qquad (50)$$

**2.3.6 Death of microbes**

The death rate of microbes is modelled as a fraction ($d_{MFT,i}$) of their active biomass (Schimel & Weintraub, 2003; Allison et
al., 2010) (Eq. (17)). Dormant microbes never die, but their biomass can be drawn to a minimal value in case of maintenance
respiration during a long period of time. The loss of C ($BAd_{C,i}$) and N ($BAd_{N,i}$) from microbial biomass due to death of
microbes is described by:

$$BAd_{X,i} = d_{MFT,i} \times BA_{X,i} \times dt \qquad (51)$$



### 2.3.7 Enzyme production and turnover

The production of enzymes is modelled as a fraction of active microbial biomass (Allison et al., 2010; He et al., 2015) depending on MFT, the saturation ratio of FOM (for enzyme *EF*) or SOM (for enzyme *ES*), and also saturation ratio of the $Avail_C$ pool. The effects of saturation ratio of substrate (FOM or SOM) and the $Avail_C$ pool on production of enzyme are

modelled using ECA kinetics (see Eq. (52), (53) and (54)). The co-effects of substrate pools and $Avail_C$ pool on enzyme production are considered following the way of Sinsabaugh and Follstad Shah (2012) considering co-limiting effects of multiple resource acquisition. Besides, a minimum amount of enzyme is produced as constitutive enzyme synthesized even under extremely unfavourable conditions (Koroljova-Skorobogat'ko et al., 1998; Kaiser et al., 2015). The production of FOM and SOM decomposing enzymes are given by Eq. (55) and (56), respectively. The deactivation of enzyme is modelled

as first order kinetics of enzyme pool (Schimel & Weintraub, 2003; Lawrence et al., 2009; Allison et al., 2010; He et al., 2015) and is given by Eq. (57) and (58) for *EF* and *ES*, respectively.

$$K_{1,FOM} = \frac{LM_C + LS_C}{KM_F \times e^{\frac{-Ea_{KM}}{R} \times \left(\frac{1}{T} - \frac{1}{T_{ref}}\right)} + LM_C + LS_C + \sum_i EF_{C,i}} \tag{52}$$

$$K_{1,SOM} = \frac{SA_C + SS_C + SP_C}{KM_S \times e^{\frac{-Ea_{KM}}{R} \times \left(\frac{1}{T} - \frac{1}{T_{ref}}\right)} + SA_C + SS_C + SP_C + \sum_i ES_{C,i}} \tag{53}$$

$$K_{2,i} = 1 - \Phi_{C,i} \tag{54}$$

$$EFg_{X,i} = BA_{X,i} \times Ke \times EFr_i \times max\left[\left(K_{1,FOM} \times K_{2,i}\right)^{\frac{1}{2}}, Ke_{min}\right] \tag{55}$$

$$ESg_{X,i} = BA_{X,i} \times Ke \times ESr_i \times max\left[\left(K_{1,SOM} \times K_{2,i}\right)^{\frac{1}{2}}, Ke_{min}\right] \tag{56}$$

$$EFd_{X,i} = EF_{X,i} \times d_{ENZ} \tag{57}$$

$$ESd_{X,i} = ES_{X,i} \times d_{ENZ} \tag{58}$$

where $EFg_{X,i}$ and $ESg_{X,i}$ are the X in newly produced enzymes *EF* and *ES* by MFT *i*, respectively; $K_{1,FOM}$ and $K_{1,SOM}$ are the

saturation ratios of FOM and SOM, respectively; $Ke \times EFr_i$ and $Ke \times ESr_i$ are the maximum enzyme production capacity for *EF* and *ES* per unit of active biomass, respectively; $Ke_{min}$ is the constitutive enzyme production constant, which is defined as fraction of maximum capacity; $d_{ENZ}$ is the turnover rate of enzymes.

### 2.3.8 Adsorption and desorption

Adsorption and desorption fluxes between the *Avail* and *Adsorb* pools are modelled as first order kinetic functions of the size

of those pools, respectively (Wang et al., 2013). Both adsorption and desorption coefficients are modulated by temperature with an activation energy of 5 ($Ea_{ads}$) and 20 ($Ea_{des}$) kJ mol[-1], respectively (Wang et al., 2013). The soil has a maximum





adsorption capacity ($Adsorb_{max}$) (Kothawala et al., 2008) because of limited mineral surface available for adsorption (Sohn and Kim, 2005). The saturation ratio of $Adsorb$ pool (defined as $Adsorb/Adsorb_{max}$) is an important factor controlling adsorption and desorption rates (Wang et al., 2013). The mass of C adsorbed ($Adsorb_{Avail,C}$) and desorbed ($Desorb_{Avail,C}$) is calculated using Eq. (59) and (60), respectively

$$Adsorb_{Avail,C} = Avail_C \times K_{ads} \times e^{-\frac{Ea_{ads}}{R} \times \left(\frac{1}{T} - \frac{1}{T_{ref}}\right)} \times \left(1 - \frac{Adsorb_C}{Adsorb_{max}}\right) \tag{59}$$

$$Desorb_{Avail,C} = K_{des} \times e^{-\frac{Ea_{des}}{R} \times \left(\frac{1}{T} - \frac{1}{T_{ref}}\right)} \times \frac{Adsorb_C}{Adsorb_{max}} \tag{60}$$

The adsorption ($Adsorb_{Avail,N}$) and desorption ($Desorb_{Avail,N}$) of N are assumed to follow the C/N ratio of the $Avail$ and $Adsorb$ pool, respectively (Eq. (61) and (62)).

$$Adsorb_{Avail,N} = Adsorb_{Avail,C} \times \frac{Avail_N}{Avail_C} \tag{61}$$

$$Desorb_{Avail,N} = Desorb_{Avail,C} \times \frac{Adsorb_N}{Adsorb_C} \tag{62}$$

Where $K_{ads}$ and $K_{des}$ are adsorption and desorption coefficients for C, respectively, and the former can be calculated from the production the latter and soil binding affinity ($K_{BA}$):

$$K_{ads} = K_{des} \times K_{BA} \tag{63}$$

**3 CENTURY and PRIM soil carbon models**

15 Here we give a brief summary of CENTURY and PRIM, the two benchmark models with which we compare ORCHIMIC for simulating incubation experiments. The CENTURY model is the SOM module of the ORCHIDEE global land biosphere model (Krinner et al., 2005). It is a simplification of the original CENTURY model (Parton et al., 1987, 1988), as it does not consider nitrogen interactions. The PRIM variant of CENTURY was developed to capture the magnitude of the priming of SOM decomposition induced by varying litter inputs (Guenet et al., 2016). Both are C-only models and have the same

20 structure with similar pools and fluxes as shown in Fig. 2. The effects of soil moisture, temperature, $pH$, lignin and clay content on decomposition of each substrate pool are also the same as those used in ORCHIMIC. Both models do not explicitly represent microbial dynamics. The decomposition rates of FOM pools in both CENTURY and PRIM (Eq. (A1) and (A2)) and the decomposition rates of SOM pools in CENTURY (Eq. (A6)-(A8)) are described by first order kinetics. The decomposition rates of the SOM pools in PRIM are modified by the size of the FOM pool and the more labile SOM

25 pools (Eq. (A6)-(A8)). The fluxes from one pool to another are exactly the same as described by Parton et al. (1987).





## 4 Parameters optimization for incubation experiments

### 4.1 Data description and model initial conditions

Data from soil incubation experiments (Blagodatskaya et al., 2014) were used to optimize the parameters of ORCHIMIC, CENTURY and PRIM using a Bayesian calibration procedure described in Sect. 4.2.

Although there are many studies investigating the priming effects of FOM addition on SOM decomposition, only few studies actually provided SOM derived respiration fluxes with and without FOM addition, as well as simultaneous FOM derived respiration fluxes and microbial biomass changes and these throughout the incubation experiment. In Blagodatskaya et al., (2014), not only were the variables mentioned above measured, also the fraction of FOM derived C in both microbial biomass and DOC was measured, which are both very useful for calibrating parameters related to microbial dynamics. As a

brief summary of their incubation experiment, [14]C labelled cellulose was added into soil as powder at a dose of 0.4 g C (kg soil) [-1] at the beginning of the incubation. The C content of the soil was 24 g C (kg soil) [-1] with a C/N ratio of 12. Soil samples with and without cellulose addition were incubated at 293.15 K at 50 % of water holding capacity for 103 days. [14]C activity and total amount of trapped $CO_2$ were measured at day 1, 4, 7, 9, 12, 14, 19, 23, 27, 33, 48, 61, 71, 90 and 103. In the meantime, microbial biomass and [14]C activity in both microbial biomass and DOC were measured at days 0, 7, 14, 60

and 103.

Nonetheless, some information required for ORCHIMIC was still not available and some assumptions were needed. The fractions of C in active, slow and passive pools were assumed to equal the fractions of C in the corresponding pools of ORCHIDEE under equilibrium at the same site where the incubated soil was sampled (Guenet et al., 2016). The C/N ratios for the three soil carbon pools were assumed equal to the ratio of total soil C and N, and the initial microbial biomass was

assumed to be equal for each MFT when more than one MFT was considered. The initial $Avail_C$ ($Avail_{C,0}$) and $Avail_N$ ($Avail_{N,0}$) pools were initialized by the initial measured DOC and DON concentration with an a priori uncertainty range of 50-150 % of the observed values. The initial ratio of active biomass ($BAr$) was set to 0.3 (ranging from 0 to 1). By assuming that the $Avail$ and $Adsorb$ pools were at equilibrium, the initial concentration of C and N in the $Adsorb$ pool ($Adsorb_{X,0}$) can be calculated from $Avail_{X,0}$ by Eq. (64). The theoretical possible maximum initial enzyme concentrations ($EF_{X,i,max}$ and

$SE_{X,i,max}$ for $EF$ and $ES$, respectively) can be estimated based on $Ke$, $EFr_i$, $ESr_i$, $d_{ENZ}$ and active microbial biomass by assuming equilibrium between active microbial biomass and enzyme concentrations, and calculated by Eq. (65) and Eq. (66), respectively. The initial enzyme concentrations for $EF$ and $ES$ is set to be any value between 0 and the theoretical possible maximum initial enzyme. $FEr$ and $SEr$, defined as the ratio of true initial enzyme concentration for $EF$ and $ES$ to their theoretical possible maximum initial enzyme concentrations, respectively, were both set to 0.1 (with a range of 0-1). The

initial concentrations for $EF$ and $ES$ are initialized as $FE_r \times FE_{X,i,max}$ and $SE_r \times SE_{X,i,max}$, respectively.

$$Adsorb_{X,0} = \frac{K_{ads} \times e^{-\frac{Ea_{ads}}{R} \times \left(\frac{1}{T} - \frac{1}{T_{ref}}\right)} \times Adsorb_{max} \times Avail_{X,0}}{K_{des} \times e^{-\frac{Ea_{des}}{R} \times \left(\frac{1}{T} - \frac{1}{T_{ref}}\right)} + K_{ads} \times e^{-\frac{Ea_{ads}}{R} \times \left(\frac{1}{T} - \frac{1}{T_{ref}}\right)} \times Avail_{X,0}}$$ (64)





$$FE_{X,i,max} = \frac{Ke \times EFr_i}{d_{ENZ}} \times B_{o,X,i} \tag{65}$$

$$SE_{X,i,max} = \frac{Ke \times ESr_i}{d_{ENZ}} \times B_{o,X,i} \tag{66}$$

where $Adsorb_{X,0}$ is the initial X (C or N) concentration in $Adsorb$ pool; $FE_{X,i,max}$ and $SE_{X,i,max}$ are theoretical maximum initial X concentrations in $EF$ and $ES$ enzyme pools, respectively; $B_{0,X,i}$ is the X in initial total microbial biomass of MFT $i$.

5  **4.2 Calibration of the parameter values in different models**

The Bayesian parameter inversion method with priors has been often used to optimize model parameters with observations (Santaren et al., 2007; Guenet et al., 2016), and was also applied in this study. The optimized parameters were determined by minimizing the following cost function $J(x)$ (Eq. (67)):

$$J(\boldsymbol{x}) = \frac{1}{2}\Big[\big(\boldsymbol{y}-\boldsymbol{H(x)}\big)^t \boldsymbol{R}^{-1}\big(\boldsymbol{y}-\boldsymbol{H(x)}\big) + (\boldsymbol{x}-\boldsymbol{x_0})^t \boldsymbol{P}^{-1}(\boldsymbol{x}-\boldsymbol{x_0})\Big] \tag{67}$$

10  where $x$ is the parameters vector for optimization; $x_0$ is the prior values vector; $P$ is the parameter error variances/covariances matrix; $y$ is the observations vector; $H(x)$ is the model outputs vector and $R$ is the observation error variances/covariances matrix. Errors are assumed to be Gaussian distributed and independent.

All parameters optimized for ORCHIMIC and their prior values and ranges are listed in Table 2. Considering that the incubation experiment was conducted at constant soil moisture, temperature and $pH$, parameters related to these variables could not be optimized and were excluded from the optimization. Also, cellulose was the only type of FOM, so $adj_{LS}$ was set to 1. The observed variables used in the optimization are listed in Table 3. All the parameters with prescribed non-optimized values are listed in Table 4. All the parameters and observed variables used in the optimization for the CENTURY and PRIM models are summarized in Tables S1 and 3, respectively. For the $R$ observation error matrix, the uncertainties of $RF$, $RS$ and $RS_{Ctrl}$ were set at 5 % of their mean observed values. Priming effect is the difference between $RS$ and $RS_{Ctrl}$, so its uncertainty was set at 10 % of the mean priming effect. The uncertainties of $B$ and $B_{Ctrl}$ were both set at 5 % of observed value. The uncertainties of $B_{FOMr}$ was set at 10 % of the observed value. The uncertainties of unknown parameters were set at 10 % of their range. The number of parameters and observations used in optimization were summarized in Table 5.

To investigate the effects of including different numbers of MFTs and also N dynamics, optimizations were performed with six variants of ORCHIMIC (C-MFT1, C-MFT2, C-MFT3, CN-MFT1, CN-MFT2 and CN-MFT3) summarized in Table 6.

25  C-only means no nitrogen dynamics are considered and the number after MFT indicates the number of MFTs used in each variant of ORCHIMIC (see details in Table 6). The gradient-based iterative algorithm L-BFGS-B (limited-memory Broyden–Fletcher–Goldfarb–Shanno algorithm) (Zhu et al.,1995) was used to minimize the cost function. As this approach may find local minima that differ from the absolute minimum of the complex function $J(x)$, it is very sensitive to the choice of initial parameter values. Guenet et al. (2016) performed 30 optimizations by assigning random initial values within a priori ranges to 6 parameters to reduce the sensitivity of the solution to the occurrence of local minima. This method was



proved to be effective to avoid potential local minima (Santaren et al., 2014). Considering the number of parameters that needed to be optimized in this model, 400 sets of random initial parameter values within their ranges were applied as initial conditions to perform optimizations for each model.

### 5 Idealized simulations increasing FOM input and/or increasing temperature

The six ORCHIMIC variants (Table 6) were forced with a constant input of 1.6 g C (kg soil)$^{-1}$ h$^{-1}$ of litter whose C/N ratio and lignin content were set to 50 and 0.2, respectively (Wang et al., 2013). In this study, only the maximum decomposition rate of cellulose was optimized, so maximum decomposition rates for C in *LM* and *LS* pools were assumed to be the same. As the temperature during the incubations was kept constant at 295.2 K, we also fixed the temperature at 295.2 K. For the CN-MFT1, CN-MFT2 and CN-MFT3 models, N was removed from the Avail pool at each time step to model the uptake of

N by vegetation. The size of the flux was chosen so that the total N flux removed from the system, including N losses during decomposition, was equal to the N input. All models were first run to equilibrium, and then three abrupt changes in the model forcings were applied, *i.e.* doubling the FOM input, increasing the temperature by 5 K, and both together.

### 6 Results

#### 6.1 Respiration and priming effect during the incubation experiment

The model simulations shown in Fig. 3 were obtained using the optimized parameters listed in Table 7 for the ORCHIMIC variants and Table S1 for the CENTURY and PRIM models. The observed respiration rate from FOM was higher at the beginning, shortly after the initial addition of labelled cellulose and gradually increased at a smaller rate. Both CENTURY and PRIM underestimated FOM derived respiration at the beginning and overestimated it at the end. Similar results were found for SOM respiration flux, with and without FOM addition (Fig. 3b and 3c). The modelled respiration from FOM and

SOM by all variants of ORCHIMIC were similar and reproduced the observed trend.

The observed cumulative priming effect, diagnosed as the difference ($RS - RS_{Ctrl}$) between $CO_2$ fluxes derived from SOM with and without FOM addition, was negative for the first 12 days and gradually became positive (Fig. 3d). Then, the cumulative priming effect increased very fast from day 14 to day 27 and after day 27, the priming effect gradually weakened. The modelled priming effect by CENTURY was always zero – by construction of this model. For PRIM, the modelled

cumulative priming effect at the end was 190 mg C (kg soil)$^{-1}$, which is 14 % higher than that observed. However, also the shape of the modelled cumulative PE curve differed from the observations. The modelled priming effect by PRIM was always positive and weakened very slowly with time (Fig. S1), making the PRIM overestimate cumulative priming effect both at the beginning and at the end, although the additional C loss through the priming effect was well captured at the end of incubation (day 103) (Fig. 3d). The negative cumulative priming effect as simulated by the various ORCHIMIC variants

lasted between 6-8 days. Similar to the observations, the modelled cumulative priming effect by the ORCHIMIC variants



increased very fast from day 8 onwards, and subsequently slowed down after 13-17 days. At the end of the experiment (day 103), the modelled cumulative priming effect by the six ORCHIMIC variants were between 170-183 mg C (kg soil)$^{-1}$, only 2.5-11 % higher than that observed.

It can be argued that ORCHIMIC does a better job at fitting the incubation data just because it has more degrees of freedom than the two other models (Table 5). The Akaike information criterion (AIC) takes this into account (Bozdogan, 1987) by considering the optimized model performance and its number of adjustable parameters. The AIC values for each model are shown in Table S2. The AIC values of the six ORCHIMIC variants are much lower than those of CENTURY and PRIM. The difference of AIC values among the six variants are very small for modelling $RF$, $RS$, $RS_{Ctrl}$ and overall performance, but C-MFT2 and CN-MFT2 have lower AIC values in modelling the priming effect.

### 6.2 Microbial biomass evolution during the incubation

Next, we examine how ORCHIMIC simulates the observed microbial biomass evolution throughout the experiment; the two other models do not explicitly include microbial biomass, so could not be evaluated here. The observed total microbial biomass increased at the beginning and reached its maximum ($\geq 442$ and $\geq 339$ mg C (kg soil)$^{-1}$ for the treatments with and without FOM addition, respectively) between day 14-60, and then decreased both with and without FOM addition (Fig. 4a). The modelled total microbial biomass from the six ORCHIMIC variants all followed a similar trend. With FOM addition, the biomass reached its maximum value of between 425-451 mg C (kg soil)$^{-1}$ on days 28-30 for the different ORCHIMIC variants. Without FOM addition, the biomass reached its maximum value of 305-325 mg C (kg soil)$^{-1}$ on days 27-34 for the different ORCHIMIC variants.

According to the observations ($^{14}$C labelling), the proportion of FOM derived C in MFT-biomass C ($B_{FOMr}$) increased very fast and peaked ($\geq 18$ %) before day 14. From day 14 to day 60, $B_{FOMr}$ declined, but subsequently increased between day 60 and day103 (Fig. 4b). The modelled $B_{FOMr}$ also increased very fast and reached its maximum value of 13-16 % on days 9-16 for the different ORCHIMIC variants. Unlike the observations, the modelled $B_{FOMr}$ continued to decrease after day 60 and declined to a value of 10-11 % for the different ORCHIMIC variants.

### 6.3 Proportion of FOM derived C in the $Avail_C$ pool during the incubation

Figure 5 shows the modelled and observed proportions of FOM derived C in the $Avail_C$ pool (defined as $Avail_{C,FOMr}$). In the observations, this quantity was not estimated as the proportion of FOM derived C in the $Avail_C$ pool, but as the proportion of FOM derived C in dissolved organic carbon (DOC). Although $Avail_C$ is not equal to DOC, we assumed that the proportion of FOM derived C in $Avail_C$ and in DOC was similar. The observed proportion of FOM derived C in DOC increased fast at the beginning and reached its maximum ($\geq 9.9$ %) before day 14 and then decreased gradually to 4.3 % on day 103. The modelled $Avail_{C,FOMr}$ reached their peaks of 29-41 % on days 2-4 for the different ORCHIMIC variants. The modelled proportion of FOM derived C in the $Avail_C$ pool on day 103 was 7-10 % for the different ORCHIMIC variants.





### 6.4 Modelled responses to step increases in temperature and fresh organic matter inputs

#### 6.4.1 Change of microbial biomass, enzymes and respiration

Figure 6 shows that at equilibrium, the standard model version CN-MFT3 of ORCHIMIC simulated a total microbial biomass of 0.17 g C (kg soil) $^{-1}$, with approximately 80 % of the microbes in the dormant and 20 % in the active state. The
total enzyme concentration was estimated to be 2.3 mg C (kg soil) $^{-1}$ and the total respiration was 3.8 mg C (kg soil) $^{-1}$ d $^{-1}$, which was equal to the C input rate. When temperature was stepwise increased by 5 K (panels a in Fig. 6), microbial biomass increased by 19 %, enzyme concentration increased by 12 %, while respiration increased much more by 42 %. However, these effects were ephemeral. After this initial peak, these three pools and fluxes declined and reached new equilibrium values, where microbial biomass was 11 % and enzyme concentrations 12 % below their original values, while
respiration rate returned to its original level equal to FOM input.

When FOM input was doubled, microbial biomass, enzyme concentration and respiration all increased and equilibrated at a higher level. Both active and dormant microbial biomass increased by 100 %, although active biomass increased faster at the beginning. Hence, the proportion of active biomass increased for about 88 days and reached a peak at 28 % (Fig. S2). Enzyme concentrations almost doubled in response to doubling FOM inputs. Respiration fluxes exactly doubled.

When both doubled FOM input and increased temperature were implemented, the temporal dynamics of microbial biomass, enzyme concentrations and respiration were very similar to those when only the FOM input doubled. At the new equilibrium, only the respiration was doubled, total microbial biomass and enzyme concentrations increased less (by 77 % and 75 %, respectively).

Although the simulated sizes of the different pools were slightly different for the other five variants of ORCHIMIC (Fig. S3-
S7), they followed similar trends as those for the standard model version.

#### 6.4.2 Change of soil carbon stock

The total SOC content, including microbial biomass and enzymes was 9.7 g C (kg soil) $^{-1}$ under equilibrium for CN-MFT3. When temperature was stepwise increased by 5 K, there was a fast decrease of C in the litter and SA pools (Fig. 7, results from the other model variants are given in Fig. S8-S12). The loss of C from the *LM* and *LS* pools reached 23 % and 26 % of
their pre-warming values, respectively. However, the decomposition rates subsequently declined and at equilibrium only 2 % of C was lost from *LS* pool and there was even a 9 % increase of C in the *LM* pool. The C stocks in the *SS* and *SP* pools decreased by 4 % and 1 %, respectively. C stocks in the *SA*, *Avail* and *Adsorb* pools decreased by 12 %, 2 % and 6 % at the new equilibrium, respectively.

With doubled FOM input, C stocks in all pools increased for a short time, but at the new equilibrium, almost did not change
(relative changes were smaller than 0.1 % for all pools after 100 years).

When both FOM input was doubled and temperature increased by 5 K, responses were almost the same as in the simulations in which only temperature was increased.





### 6.4.3 Changes of carbon use efficiency

At equilibrium, the carbon use efficiency (CUE), defined as ratio of carbon allocated to microbial growth to the sum of those allocated to growth and respiration, was between 0.40-0.44 for the different ORCHIMIC variants. When T was increased by 5 K, CUE first fluctuated but finally stabilized at slightly lower values (between 0.39-0.42) in all ORCHIMIC variants (Fig.
8a).

When FOM input was doubled, CUE transiently increased for 52-73 days to a maximum value between 0.46-0.49 for the different ORCHIMIC variants. At the new equilibrium, however, CUE was similar to its original level in all ORCHIMIC variants (Fig. 8b).

When T was stepwise increased by 5K and FOM input doubled, CUE responses were in between those to warming and those
to increased FOM additions. At equilibrium, however, the CUE response was similar to that in the T-only treatment for all ORCHIMIC variants (Fig. 8c).

### 7 Discussion

### 7.1 Optimized parameter vs. literature values

The optimized values for most parameters were generally consistent with those used by previous models and those observed.
For example, the ratios of the decomposition rates of the active to slow SOC pool and of the slow to the passive pool were close to those used in the original CENTURY model (Parton et al., 1987). The optimized turnover rate of enzymes (0.035-0.065 $d^{-1}$ for the six ORCHIMIC variants) was within the range of observed turnover rates for enzymes (0.002-0.10 $d^{-1}$) (Schimel et al., 2017) and also of similar magnitude than those used in the models of Allison et al. (0.024 $d^{-1}$) (2010), He et al. (0.012-0.048 $d^{-1}$) (2015), Schimel and Weintraub (0.05 $d^{-1}$) (2003) and Lawrence et al. (0.05 $d^{-1}$) (2009). The optimized
maximum C uptake rate of microbes (0.29-0.74 $h^{-1}$) was higher, but nonetheless of the same order of magnitude than the value of 0.24 $h^{-1}$ used by Allison et al (2010), although much higher than the value of 0.0005 $h^{-1}$ used by Wang et al. (2013). The optimized value of death rate of active microbes (0.0015-0.0027 $h^{-1}$) was consistent with observations. For example, the measured death rate for total microbial biomass at 298.15 K was 0.016 $d^{-1}$ (Joergensen et al., 1990). Hence, considering an active biomass proportion of 4-49 %, the death rate for active biomass would be 0.0014-0.017 $h^{-1}$. The optimized death rate
for active microbial biomass was also consistent with those used for active biomass by He et al. (0.0002-0.002$h^{-1}$; 2015), and comparable to those used by Allison et al. (2010) and Lawrence et al. (2009) (0.0002 and 0.0021 $h^{-1}$for total microbial biomass, respectively) if considering an active biomass proportion of 4-49 % (Van de Werf and Verstraete, 1987). Other optimized parameter values that were directly comparable to observations were also consistent with empirical data.  For example, the proportion (0.0042-0.0053) of initial $Avail_C$ in total SOC was close to the value of 0.0041 for the proportion of
DOC in total SOC reported by Blagodatskaya et al. (2014) for the incubated soil. The initial active microbial biomass




proportion was 26- 48 % of the total biomass, lying in the observed range of 4-49 % reported by Van de Werf and Verstraete (1987).

Some other optimized parameters differed substantially from the values used in previous models, yet were consistent with those observed. For example, the ratio of maintenance respiration in dormant relative to active microbes (0.12-0.24) was

within the range reported by Wang et al. (2014) (0.025-0.351) estimated based on data from two incubation experiments, but much higher than that used in the model of He et al (2015) (0.0005-0.005). The optimized CAE of 0.8 was also higher than the value of 0.5 used by Schimel and Weintraub (2003), yet close to the value (0.8) for CAE of reserve metabolites used in the model of Tang and Riley (2015). A wide range of CUE (0.01-0.85) was reported by Six et al. (2006) in a review of studies measuring CUE. High CUE (0.67-0.75) was also reported by Hagerty et al. (2014). These high values indicate that

CAE could be as high as 0.8 because CAE should be larger than CAE as CUE takes maintenance respiration into account. The maximum decomposition rates of substrates were higher than those used in previous models (Allison et al., 2010; Wang et al., 2013; Kaiser et al., 2014, 2015). For example, in Wang et al. (2013), the optimized maximum decomposition rates for particulate organic matter and mineral-associated organic matter were 2.5 and 1.0 mg C (mg enzyme C) $^{-1}$ h$^{-1}$, respectively, and 0.24 mg C (mg enzyme C) $^{-1}$ h$^{-1}$ was used as maximum decomposition rate for soil organic matter in the model of

Allison et al. (2010). However, the maximum decomposition rate for cellulose optimized from our study was 83-190 mg C (mg enzyme C) $^{-1}$ h$^{-1}$. One likely explanation for such a large difference is that the data used by Wang et al., (2013) and Allison et al. (2010) were aimed at being applied to decomposition of SOM or litter, while in this study, the main substrate was cellulose which was milled before being added to soil and was also well mixed within the soil during the incubation experiment. Moreover, cellulose has a very homogeneous structure and is therefore easy to decompose. Anyway, the

maximum decomposition rate is within the range reported by laboratory measurements for cellulose. For example, according to data collected by Wang et al. (2012), the maximum decomposition rate of cellulose could be as high as 7900 mg C (mg enzyme C) $^{-1}$ h$^{-1}$ with an average of 80 mg C (mg enzyme C) $^{-1}$ h$^{-1}$.

### 7.2 Performance of ORCHIMIC model

The ORCHIMIC model generally performed better than CENTURY and PRIM. Despite the larger number of parameters, the

AIC values for the six variants of ORCHIMIC were lower than those of the more parsimonious CENTURY and PRIM models. The decomposition rates in CENTURY follow first order kinetics (Parton et al., 1987) and do not interact, so with and without FOM addition, the SOM derived respiration is always the same and priming cannot be captured. The PRIM model was developed with the aim of modelling the priming effect (Guenet et al., 2016). The decomposition rate of FOM still follows first order kinetics, so FOM derived respiration has a similar trend as in the CENTURY model. However, the

decomposition rate of more recalcitrant SOC is accelerated when the FOM pool is higher, as is the case in incubations with FOM (cellulose) addition. Hence, SOM derived respiration will increase and lead to a positive priming effect of rather constant magnitude for the simulations where cellulose is added. In contrast, the ORCHIMIC with different numbers of



MFTs and with or without N dynamics all better captured the temporal dynamics of both respiration and priming effects measured by Blagodatskaya et al (2014).

In ORCHIMIC, the substrate decomposition rate is non-linear because ECA kinetics are applied in simulating substrate decomposition (Eq. (20)-(24)). The decomposition rate becomes lower as the substrate gradually depletes (Fig. 3a) because
the incubation experiments do not have continuous input of C like in the real world. This model result is consistent with observations of decelerating respiration at the end of the incubation (Fig. 3a). In ORCHIMIC, the depletion of substrates lowers the saturation ratio of the substrate pool and subsequently inhibits the production of enzymes and reduces the decomposition rate of the substrates. The resulting lower saturation ratio of the *Avail* pool then triggers dormancy and reduces the growth rate of active microbial biomass, which in turn reduces enzyme production and thereby generates a
positive feedback to reduced decomposition. As a result, SOM mineralization rates and respiration rates slow down at the end of the incubation experiment.

The main mechanism underlying the positive priming effect in ORCHIMIC is that the FOM input stimulates the growth of active microbes and transformation of dormant states to active states, leading to increased enzyme production and thereby faster mineralization of SOM. However, at the beginning, the fast mineralization of FOM decreases the fraction of SOM
derived C in the *Avail* pool. The total respiration does not change much, but less respired C is SOM derived, thus creating a negative priming effect. Also, because of applying dynamic enzyme production, the increase of the saturation ratio of the *Avail* pool due to FOM addition suppresses enzyme production per unit of active biomass, and thus slows down the increase of, or even decreases, the size of the SOM-decomposition enzyme pool, which partly suppresses SOM derived respiration.

ORCHIMIC reproduced the observed microbial biomass, a variable which is not modelled by CENTURY and PRIM. Also,
the transfer of FOM derived C to the *Avail* pool and the assimilation of FOM derived C into microbial biomass were well captured (Fig. 4b and 5). However, the observed increased contribution of FOM derived C in microbial biomass during the incubation was not reproduced by ORCHIMIC. This suggests that some important processes related to microbial biomass are misrepresented or still lacking. As there was no such increase for DOC, the increase of the proportion of FOM derived C in microbial biomass was probably not due to the increased uptake of FOM derived C. Therefore, this is probably related to
microbial turnover, which is homogeneous for old (more is SOM derived) and new (more is FOM derived) microbial biomass C in ORCHIMIC.

With the same FOM input but under a lower temperature 285.15 K, Wang et al. (2013) simulated a SOC stock of about 17 g C (kg soil)$^{-1}$, which was 2-3 times of that simulated here. This may be attributable to the much smaller decomposition rates applied in their model. In our study, the equilibrium C concentration in the *Avail* pool was 0.11-0.32 g C (kg soil)$^{-1}$,
comparable with 0.16 g C (kg soil)$^{-1}$ in their model for dissolved organic carbon (DOC), and within one standard deviation interval of the range 0.04-0.52 g C (kg soil)$^{-1}$ reported in the literature (Wang et al., 2013). In ORCHIMIC, the total enzyme concentration at equilibrium was 1.78-5.75 mg C (kg soil)$^{-1}$, which was close to the reported upper range (0.01-5 mg C (kg soil)$^{-1}$) for α-glucosidase and β-glucosidase concentrations in soil by Tabatabai, (2003). However, considering that many kinds of enzymes exist in soil, Wang et al. (2003) used a value of 1 mg C (kg soil)$^{-1}$ when estimating parameter values for



their model. Hence, the enzyme concentrations simulated by ORCHIMIC are probably realistic. ORCHIMIC generated a reasonable proportion of microbial biomass in the total soil C stock (1.8-4.4 %), which is around the global average of in-situ measurements compiled by Xu et al. (2013). The active biomass proportion was also close to that reported by Van de Werf and Verstraete (1987) (19±9 %), by Lennon and Jones (2011) (18±15 %), and by Stenström et al. (2001) (5-20 %).

All soil C pools except *LM*, decreased in response to warming, which was consistent with the simulations by the conventional SOM decomposition models. Unlike other pools, there was an increase for the *LM* pool, because the increase of decomposition rate per unit of enzymes was relative small due to the lower temperature sensitivity of the decomposition of LM (prescribed smallest *Ea* for *LM* in ORCHIMIC) and it was compensated by the decreased enzyme concentration. The soil C pools almost did not change with increased FOM inputs, because increasing FOM accelerated the decomposition of

SOM by stimulating the growth of microbes and their production of enzymes. These responses were totally different from the proportional increase in soil C pools as modelled by the conventional linear SOC decomposition model. With double FOM input and warming, the modelled SOC stock by ORCHIMIC decreased instead of increased as modelled by the conventional linear SOC decomposition model. This was because the priming effect induced by FOM addition compensated the increased C input to the soil and because the increased SOC decomposition rate due to warming decreased the SOC

stock.

### 7.3 Comparison of 6 ORCHIMIC variants

Regarding the simulation of respiration, the priming effect and microbial biomass, the cost function value at the minimum was the smallest for the two ORCHIMIC model variants with two MFTs and largest for the more comprehensive standard version CN-MFT3 (Table 7). Thus, any improvements associated with having more MFTs or including N dynamics in the

model are not apparent when using the Blagodatskaya et al. (2014) measurements. In ORCHIMIC, the main differences among the MFTs are their different C/N ratio and the ability to produce two kinds of enzyme. Although ORCHIMIC can simulate different enzyme concentrations with different MFTs, their effects were partly offset by the different maximum decomposition rates for the different enzymes, making models with more MFTs not always better than models with fewer MFTs. Also, the assumption that the initial biomass and active biomass proportion of each MFT limited the performance of

the model with more MFTs.

According to our simulations for the different temperature and FOM-addition scenarios, the amount of N required for microbial growth was only 10 % of the initial DON in the incubated soil. Therefore, N was sufficiently available to feed microbial demand, explaining why the model set ups without the N cycle behaved similar to those with the representation of the N cycle. Future applications of this model, using N-limited soils are needed to assess to what degree N cycling needs to

be represented in the SOC models. Because the long-term limitation of N on microbial growth was absent from our study, we cannot yet evaluate the potential improvements by including N dynamics in the model.

When N is considered, N-limited conditions favor the growth of MFTs with larger C/N ratios, while C-limited conditions favor the growth of MFTs with smaller C/N ratios. Thus, with more than one MFT, the C/N ratio of total microbial biomass





is variable, according to the C/N ratio of the *Avail* pool that is calculated from the C/N ratio of the substrate and the decomposition ability of each type of MFT.

### 7.4 Carbon use efficiency changes with warming and increased FOM input

As model suggested that, upon a 5K stepwise increase of temperature, CUE initially decreased by 0.05-0.08 due to the
immediate increase of maintenance respiration in response to the higher temperature. However, at equilibrium, the change in temperature induced a decrease of CUE by 0.0018-0.0026 K$^{-1}$, relative to the equilibrium at lower temperature. The activation energy is a key factor regulating the response of the maintenance respiration cost to warming. In this study, the activation energy for maintenance respiration was set to 20 kJ mol$^{-1}$ (van Iersel and Seymour, 2002), while 60 kJ mol$^{-1}$ was used by Tang and Riley (2015). A larger activation energy implies lower respiration rates, but also a larger temperature
sensitivity of respiration, and thus a larger relative increase of the maintenance cost and a larger decrease of CUE when temperature increases (Davidson and Janssens, 2006; Davidson et al., 2012). Although not always consistent among experimental studies (Dijkstra et al., 2011), a decrease of CUE with warming has often been observed. For example, Van Ginkel et al. (2000) showed that the sensitivity of CUE in response to warming could be as large as -0.049 K$^{-1}$ and Steinweg et al. (2008) found CUE sensitivity to warming of -0.009 K$^{-1}$. After considering the effect of temperature on the turnover of
microbial biomass, Hagerty et al. (2014) estimated a decrease of CUE by 0.005 and 0.003 K$^{-1}$ for mineral and organic soil, respectively. There are indications that the temperature response of CUE varies with substrate and temperature. For example, in the study of Devêvre and Horwath (2000), when temperature increased from 278.15 K to 288.15 K, the CUE decreased by 0.021 and 0.015 K$^{-1}$ when the soil was incubated with low C/N and high C/N of FOM, respectively. That study also showed that, when temperature increased further from 288.15 K to 298.15 K, the decrease of CUE was only 0.006 K$^{-1}$. It thus seems
that CUE tends to decrease more slowly when the applied temperature warming or the C/N ratio of FOM is higher. If this is true, then the relatively low decrease in CUE of 0.0018-0.0026 K$^{-1}$ that we observed was to be expected.

Besides temperature and the substrate's C/N ratio, the decomposing rate of the substrate is also an important factor affecting CUE. Normally, CUE is larger with substrates of higher decomposing rate, because the maintenance costs remain relatively stable (del Giorgio and Cole, 1998, van Bodegom, 2007). As such, the short-lived increase of CUE that we observed in Fig.
8 after temperature was increased by 5 K may be related to the increase of the maximum decomposition rate for each substrate with temperature. Also, with doubled FOM input, a longer increase of CUE compared to that with the 5 K stepwise increase of temperature was found. However, at the new equilibrium, with doubled FOM inputs, the respiration doubled because microbial biomass also doubled, and therefore the CUE remained almost unchanged.

## 7.5 Implications

Increased greenhouse gases in the atmosphere warm air temperature, and subsequently soil temperature, which increases primary productivity in regions where water or nutrients are not scarce. Thus, increases of both input and decomposition





rates of SOM are expected (Jones et al. 2005). The response of SOC to these two drivers (input and decomposition rates) is determined by complex processes, but current SOC decomposition models used in Earth System Models (ESM) always simulate that increased input leads to increased storage of SOC, and that soil warming leads to decreased storage from increased decomposition rates. Despite different pathways of $CO_2$ emission scenarios, SOC stocks tends to increase in the near future in most ESMs (Burke et al., 2017; Todd-Brown et al., 2014). In ORCHIMIC, as shown in Fig. 7 for the 5 K and doubled FOM input simulations, the canonical model response is that SOC stocks are projected to decrease instead of increase, implying a totally different response of SOC stock to future climate change as projected by conventional linear SOC decomposition models.

The performance ORCHIMIC variants CN-MFT3 does not obviously perform better than the variants with less MFTs and even those without N dynamics in reproducing the results from incubation experiments. However, CN-MFT3, as the standard version of ORCHIMIC, is preferable as it is able to model dynamic C/N ratio of microbial community and also is more accurate in modelling dynamics of soil SOM pools including microbial biomass pools under N-limited conditions.

Despite the complexity of ORCHIMIC compared to the current SOC models embedded in ESMs for large scale applications, the main soil carbon and litter pools in the model are defined similarly as those of ESMs, and most of the input variables like litter fall and plant N uptake, and environmental conditions (soil moisture and temperature) can also be directly calculated by the ESM. Also, the time step of ORCHIMIC is similar to that of most ESMs (*i.e.* daily). This makes it possible to embed ORCHIMIC into most current ESMs. As ORCHIMIC includes key processes related to microbial communities that can be measured in experiments, it provides the basis for a refined representation of global change effects on soil C by integrating intertwined processes such as soil nutrient availability and organic matter inputs.

## 8 Conclusions

We developed a soil C and N model with a dynamic enzyme production mechanism and a microbial dormancy strategy considered for four microbial function groups, with generalists, FOM specialists and SOM specialists explicitly represented and cheaters inexplicitly included. This newly developed ORCHIMIC model not only reproduces respiration, but also the priming effect. Moreover, it can reproduce several measurable variables, such as microbial biomass, not only the total microbial biomass, but also the fractions of active microbial biomass and SOM and FOM derived C in the total microbial biomass. In addition, with realistic inputs, ORCHIMIC generated realistic SOC stocks, microbial biomass, proportion of microbial biomass in the SOC stock, proportion of active microbial biomass in total microbial biomass, as well as enzyme concentrations. Finally, ORCHIMIC can be easily integrated into ESMs for more realistic predictions of changes in SOM under future scenarios.

## Code and data availability




The ORCHIMIC v1.0 is programed in Python language and the run of model need Python with basic packages (numpy, os and sys) preinstalled. The source code, optimized parameter values and script used to reproduce the results showed in Sect. 6.4 are available online (https://github.com/huangysmile/ORCHIMIC/releases/tag/v1.0; DOI: 10.5281/zenodo.1164740).

**Competing interests**

The authors declare that they have no conflict of interest.

**Acknowledgements**

Funding for the study was supported by the European Research Council through Synergy grant ERC-2013-SyG610028 "IMBALANCE-P".

**Appendix A: equations describing dynamics of pools for CENTURY and PRIM models**

$$D_{LM} = K_{LM} \times LM \times F_\theta \times F_{T,LM} \times F_{pH} \tag{A1}$$

$$D_{LS} = \frac{K_{LM}}{Adj_{LS}} \times LS \times F_\theta \times F_{T,LS} \times F_{pH} \times F_{lignin} \tag{A2}$$

$$D_{SA,CENT} = K_{SS} \times Adj_{SA} \times SA \times F_\theta \times F_{T,SA} \times F_{pH} \times F_{clay} \tag{A3}$$

$$D_{SS,CENT} = K_{SS} \times SS \times F_\theta \times F_{T,SS} \times F_{pH} \tag{A4}$$

$$D_{SP,CENT} = \frac{K_{SS}}{Adj_{SP}} \times SP \times F_\theta \times F_{T,SP} \times F_{pH} \tag{A5}$$

$$D_{SA,PRIM} = K_{SS} \times Adj_{SA} \times SA \times F_\theta \times F_{T,SA} \times F_{pH} \times F_{clay} \times \left[1 - e^{-c_{SA} \times (LM+LS)}\right] \tag{A6}$$

$$D_{SS,PRIM} = K_{SS} \times SS \times F_\theta \times F_{T,SS} \times F_{pH} \times \left[1 - e^{-c_{SS} \times (LM+LS+SA)}\right] \tag{A7}$$

$$D_{SP,PRIM} = \frac{K_{SS}}{Adj_{SP}} \times SP \times F_\theta \times F_{T,SP} \times F_{pH} \times \left[1 - e^{-c_{SP} \times (LM+LS+SA+SS)}\right] \tag{A8}$$

where $F_\theta$, $F_{T,j}$, $F_{pH}$, $F_{clay}$ and $F_{lignin}$ are functions of soil moisture, temperature, pH, clay content, lignin content with the same definitions in ORCHIMIC; $Adj_{LS}$, $Adj_{SA}$ and $Adj_{SP}$ are also defined same as in ORCHIMIC; $K_{LM}$ and $K_{SS}$ are decomposition

rate of C in $LM$ and $SS$ pools; $D_{SA,CENT}$, $D_{SS,CENT}$ and $D_{SP,CENT}$ are decomposition fluxes of C for $SA$, $SS$ and $SP$ pools in CENTURY; $D_{SA,PRIM}$, $D_{SS,PRIM}$ and $D_{SP,PRIM}$ are decomposition fluxes of C for $SA$, $SS$ and $SP$ pools in PRIM.

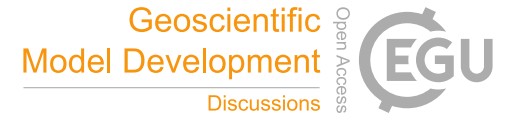

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



**Table 1. List of variables and parameters used in this study.**

| Variables used in the ORCHIMIC | | |
| --- | --- | --- |
| Variables | Description | Units |
| $dt$ | Time step | 24 h |
| $i$ | Represents Microbial Functional Type $i$ | [1, 2, 3] |
| $j$ | Represents substrate $j$ | [LM, LS, SA, SS, SP] |
| X | Represents C or N element | [C, N] |
| $LM_X$ | X in litter metabolic pool | g X (kg soil)$^{-1}$ |
| $LS_X$ | X in litter structural pool | g X (kg soil)$^{-1}$ |
| $SA_X$ | Soil active organic matter pool | g X (kg soil)$^{-1}$ |
| $SS_X$ | Soil slow organic matter pool | g X (kg soil)$^{-1}$ |
| $SP_X$ | Soil passive organic matter pool | g X (kg soil)$^{-1}$ |
| $LLf$ | Lignin fraction of the LS pool | unitless |
| $Avail_X$ | X pool directly available for microbe's uptake | g X (kg soil)$^{-1}$ |
| $Adsorb_X$ | X pool adsorbed on mineral surfaces | g X (kg soil)$^{-1}$ |
| $BA_{X,i}$ | X in active microbial biomass of MFT $i$ | g X (kg soil)$^{-1}$ |
| $BD_{X,i}$ | X in dormant microbial biomass of MFT $i$ | g X (kg soil)$^{-1}$ |
| $EF_{X,i}$ | X in enzyme produced by MFT $i$ that can decompose FOM | g X (kg soil)$^{-1}$ |
| $ES_{X,i}$ | X in enzyme produced by MFT $i$ that can decompose SOM | g X (kg soil)$^{-1}$ |
| FOM$_X$ | Fresh organic matter pools for X ($LM_X+LS_X$) | g X (kg soil)$^{-1}$ |
| SOM$_X$ | Soil organic matter pools for X ($SA_X+SS_X+SP_X$) | g X (kg soil)$^{-1}$ |
| $LM_{X,in}$ | Input of X for $LM$ | g X (kg soil)$^{-1}$ $dt^{-1}$ |
| $LS_{X,in}$ | Input of X for $LS$ | g X (kg soil)$^{-1}$ $dt^{-1}$ |
| $EFg_{X,i}$ | X in new $EF_i$ produced in one time step | g X (kg soil)$^{-1}$ $dt^{-1}$ |
| $ESg_{X,i}$ | X in new $ES_i$ produced in one time step | g X (kg soil)$^{-1}$ $dt^{-1}$ |
| $EFd_{X,i}$ | X in $EF_i$ that is deactivated in one time step | g X (kg soil)$^{-1}$ $dt^{-1}$ |
| $ESd_{X,i}$ | X in $ES_i$ that is deactivated in one time step | g X (kg soil)$^{-1}$ $dt^{-1}$ |
| $BAg_{X,i}$ | X in new $BA_i$ produced in one time step | g X (kg soil)$^{-1}$ $dt^{-1}$ |
| $BAd_{X,i}$ | X in $BA_i$ that died in one time step | g X (kg soil)$^{-1}$ $dt^{-1}$ |
| $BAm_{X,i}$ | $BA_{X,i}$ lost due to maintenance respiration in one time step | g X (kg soil)$^{-1}$ $dt^{-1}$ |
| $BDm_{X,i}$ | $BD_{X,i}$ lost due to maintenance respiration in one time step | g X (kg soil)$^{-1}$ $dt^{-1}$ |
| $B_{AtoD,X,i}$ | X transformed from $BA_{X,i}$ to $BD_{X,i}$ in one time step | g X (kg soil)$^{-1}$ $dt^{-1}$ |



| | | |
|---|---|---|
| $B_{DtoA,X,i}$ | X transformed from $BD_{X,i}$ to $BA_{X,i}$ in one time step | g X (kg soil)$^{-1}$ $dt^{-1}$ |
| $Adsorb_{Avail,X}$ | Adsorbed $Avail_X$ in one time step | g X (kg soil)$^{-1}$ $dt^{-1}$ |
| $Desorb_{Adsorb,X}$ | Desorbed $Adsorb_X$ in one time step | g X (kg soil)$^{-1}$ $dt^{-1}$ |
| $Uptake_{X,i}$ | Uptake of X by MFT $i$ in one time step | g X (kg soil)$^{-1}$ $dt^{-1}$ |
| $Uptakeadj_{X,i}$ | Adjusted $Uptake_{X,i}$ | g X (kg soil)$^{-1}$ $dt^{-1}$ |
| $g_{C,i}$ | Growth rate if only consider C for MFT $i$ | g X (kg soil)$^{-1}$ $dt^{-1}$ |
| $g_{N,i}$ | Growth rate if only consider N for MFT $i$ | g X (kg soil)$^{-1}$ $dt^{-1}$ |
| $Rg_i$ | Growth respiration of MFT $i$ | g X (kg soil)$^{-1}$ $dt^{-1}$ |
| $Ro_i$ | Overflow respiration of MFT $i$ | g X (kg soil)$^{-1}$ $dt^{-1}$ |
| $RAm_j$ | Maintenance respiration of active MFT $i$ | g X (kg soil)$^{-1}$ $dt^{-1}$ |
| $RDm_i$ | Maintenance respiration of dormant MFT $i$ | g X (kg soil)$^{-1}$ $dt^{-1}$ |
| $Rm_i$ | $RAm_i + RDm_i$ | g X (kg soil)$^{-1}$ $dt^{-1}$ |
| $D_{X,j}$ | Flux of X decomposed from substrate $j$ | g X (kg soil)$^{-1}$ $dt^{-1}$ |
| $Dloss_{N,j}$ | Gaseous N losses during decomposition of substrate $j$ | g X (kg soil)$^{-1}$ $dt^{-1}$ |
| $Veg_{uptake,N}$ | N uptake by vegetation | g X (kg soil)$^{-1}$ $dt^{-1}$ |

Parameters used in ORCHIMIC

| Parameters | Description | Units |
|---|---|---|
| $LLf_{in}$ | Lignin fraction of input litter | unitless |
| $LCN_{in}$ | C/N mass ratio of input litter | unitless |
| $LMf$ | Fraction of input litter allocated to $LM$ | unitless |
| $T_{ref}$ | Reference temperature | K |
| $T$ | Soil temperature | K |
| $\theta$ | Soil moisture: fraction of field capacity [0-1] | unitless |
| $pH$ | Soil $pH$ | pH units |
| $pH_{0,ENZ}$ | Optimum $pH$ for decomposing substrate | pH units |
| $pH_{s,ENZ}$ | Sensitivity parameter to $pH$ for decomposing substrate | pH units |
| $\theta_{0,i}$ | Optimum $\theta$ for growth of MFT i | unitless |
| $pH_{0,i}$ | Optimum $pH$ for growth of MFT i | pH units |
| $\theta_{s,i}$ | Growth sensitivity parameter to $\theta$ for MFT $i$ | unitless |
| $pH_{s,i}$ | Growth sensitivity parameter to $pH$ for MFT $i$ | pH units |
| $LtoSS$ | The fraction of decomposed $LM$ and non-lignin $LS$ that go to SS pool | unitless |
| $SAtoSS$ | The fraction of decomposed $SA$ that go to $SS$ pool | unitless |
| $SAtoSP$ | The fraction of decomposed $SA$ that go to $SP$ pool | unitless |





| | | |
|---|---|---|
| $SStoSP$ | The fraction of decomposed $SS$ that go to $SP$ pool | unitless |
| $BCN_i$ | C/N ratio for MFT $i$ | unitless |
| $CC$ | Soil clay content | unitless |
| $Vmax_{uptake,i}$ | Maximum uptake rate of C at optimum conditions for MFT $i$ | $h^{-1}$ |
| $Ke$ | Maximum enzyme production coefficient | $h^{-1}$ |
| $EFr_i$ | Maximum FOM decomposing enzyme production capacity of MFT $i$ | unitless |
| $ESr_i$ | Maximum SOM decomposing enzyme production capacity of MFT $i$ | unitless |
| $Kr_{ref}$ | Maintenance respiration coefficient of microbes at $T_{ref}$ | $h^{-1}$ |
| $Kr$ | Maintenance respiration coefficient of microbes at $T$ | $h^{-1}$ |
| $b$ | Ratio of maintenance respiration rate for $BD$ to $BA$ | unitless |
| $d_{MFT,i}$ | Death rate of MFT $i$ | $h^{-1}$ |
| $d_{ENZ}$ | Turnover rate of enzymes | $h^{-1}$ |
| $R$ | Ideal gas constant, 0.008314 | $kJ\ mol^{-1}\ K^{-1}$ |
| $Ea_{main}$ | Activation energy for maintenance respiration | $kJ\ mol^{-1}$ |
| $Ea_j$ | Activation energy for decomposition of substrate $j$ | $kJ\ mol^{-1}$ |
| $Vmax_j$ | Maximum decomposition rate for substrate j at $T_{ref}$ | $g\ C\ (g\ ENZ\ C)^{-1}\ h^{-1}$ |
| $Adj_{LS}$ | Ratio of decomposition rate of $LM$ to that of $LS$ | Unitless |
| $Adj_{SA}$ | Ratio of decomposition rate of $SA$ to that of $SS$ | Unitless |
| $Adj_{SP}$ | Ratio of decomposition rate of $SS$ to that of $SP$ | unitless |
| $KM_F$ | Michaelis-Menten constant for decomposition of FOM | $g\ C\ (kg\ soil)^{-1}$ |
| $KM_S$ | Michaelis-Menten constant for decomposition of SOM | $g\ C\ (kg\ soil)^{-1}$ |
| $Ea_{KM}$ | Activation energy for Michaelis-Menten constants | $kJ\ mol^{-1}$ |
| CAE | Carbon assimilation efficiency | unitless |
| NAE | Nitrogen assimilation efficiency | unitless |
| $s_C$ | Soluble fraction of dead microbial for C | unitless |
| $s_N$ | Soluble fraction of dead microbial for N | unitless |
| $K_{ads}$ | Avail pool adsorption coefficient at $T_{ref}$ | $h^{-1}$ |
| $K_{des}$ | Adsorb pool desorption coefficient at $T_{ref}$ | $h^{-1}$ |
| $Adsorb_{max}$ | Max adsorption capacity of soil | $g\ C\ (kg\ soil)^{-1}$ |
| $K_{BA}$ | Soil binding affinity, $K_{ads}/K_{des}$ | unitless |
| $TAvail_X$ | Total available X considering those from decomposition and dead microbes | $g\ X\ (kg\ soil)^{-1}$ |
| $KM_{uptake,X,i}$ | Michaelis-Menton constant for uptake of X for MFT $i$ | $g\ X\ (kg\ soil)^{-1}$ |
| $Ea_{uptake}$ | Activation energy for uptake | $kJ\ mol^{-1}$ |



| | | |
|---|---|---|
| $\Phi_{C,i}$ | Saturation ratio of directly available organic C for MFT $i$ | unitless |
| $Ke_{min}$ | Minimum (or constitutive) enzyme production coefficient, defined as ratio of maximum capacity | unitless |
| $Availr$ | Ratio of C in $Avail$ pool to total soil C at beginning | unitless |
| $FEr$ | Parameter for initial total $EF$ concentration | unitless |
| $SEr$ | Parameter for initial total $ES$ concentration | unitless |
| $BAr$ | Initial active biomass ratio | unitless |
| $Adsorb_{X,0}$ | Initial X (C or N) concentration in $Adsorb$ pool | g X (kg soil)$^{-1}$ |
| $FE_{X,i,max}$ | Theoretical maximum initial X concentrations in $EF$ enzyme pools | g X (kg soil)$^{-1}$ |
| $SE_{X,i,max}$ | Theoretical maximum initial X concentrations in $ES$ enzyme pools | g X (kg soil)$^{-1}$ |
| $B_{0,i}$ | Initial total microbial biomass for MFT $i$ | g C (kg soil)$^{-1}$ |
| $K_j$ | Decomposition coefficient of substrate $j$ in CENTURY or RPIM model | $dt^{-1}$ |
| $c_{SA}, c_{SS}, c_{SP}$ | Priming parameters for decomposition of $SA$, $SS$ and $SP$ for PRIM, respectively | kg soil (g C)$^{-1}$ |

**Table 2. List of optimized parameters with their prior values and ranges; for the meaning of each parameter see Table 1.**

| Parameters | Units | Prior values | Ranges | References |
|---|---|---|---|---|
| $Adj_{SA}$ | unitless | 37 | 32-42 | Parton et al., 1987 |
| $Adj_{SP}$ | unitless | 29 | 24-34 | Parton et al., 1987 |
| $AvailCr$ | unitless | 0.0041 | 0.0021-0.0061 | Blagodatskaya et al., 2014; Wang et al., 2013 |
| $b$ | unitless | 0.01 | 0.0005-1 | He et al., 2015; Wang et al., 2014 |
| $BAr$ | unitless | 0.3 | 0-1 | Wang et al., 2014 |
| CAE | unitless | 0.6 | 0.01-0.85 | Schimel and Weintraub, 2003; Six et al., 2006 |
| $d_{ENZ}$ | h$^{-1}$ | 0.001 | 0.0005-0.016 | Allison et al. 2010 ; Kaiser et al., 2014, 2015; He et al., 2015 |
| $d_{MFT}$ | h$^{-1}$ | 0.002 | 0.0002-0.01 | Allison et al., 2010; He et al., 2015; Kaiser et al., 2014 |
| $FEr$ | unitless | 0.1 | 0.00001-1 | This study |
| $K_{BA}$ | unitless | 6 | 1-11 | Wang et al., 2013 |
| $K_{des}$ | h$^{-1}$ | 0.001 | 0.0001-0.01 | Wang et al., 2013 |
| $Ke$ | h$^{-1}$ | 0.00001 | 0.000005-0.0008 | Allison et al., 2010 He et al., 2015 |
| $Kr_{ref}$ | h$^{-1}$ | 0.002 | 0.0001-0.08 | Kaiser et al., 2014; He et al., 2015 |



| | | | | |
|---|---|---|---|---|
| $KM_F$ | gC (kg soil)$^{-1}$ | 50 | 0.01-100 | Wang et al., 2013; Allison et al., 2010; He et al., 2015 |
| $KM_S$ | gC (kg soil)$^{-1}$ | 250 | 0.01-500 | Wang et al., 2013; Allison et al., 2010; He et al., 2015 |
| $KM_{uptake}$ | gC (kg soil)$^{-1}$ | 0.26 | 0.0026-26 | Wang et al., 2013; Allison et al., 2010 |
| $LtoSS$ | unitless | 0.02 | 0-0.5 | Wieder et al., 2014; D'Odorico et al., 2003 |
| $SEr$ | unitless | 0.1 | 0.00001-1 | This study |
| $Adsorb_{max}$ | g C (kg soil)$^{-1}$ | 1.35 | 0.5-4.8 | Mayes et al., 2012 |
| $Vmax_{LM}$ | g C (g ENZ C)$^{-1}$h$^{-1}$ | 56 | 7-447 | Wang et al., 2012 |
| $Vmax_{SS}$ | g C (g ENZ C)$^{-1}$h$^{-1}$ | 1 | 0.008-50 | Wang et al., 2012; Wang et al., 2013 |
| $Vmax_{uptake,C}$ | g C (g ENZ C)$^{-1}$h$^{-1}$ | 0.24 | 0.0005-2 | Wang et al., 2013; Zwietering et al.1991; Weiger et al., 1995 |

**Table 3. List of the observed variables used for optimization.**

| Variables | Units | Descriptions | ORCHIMIC | CENTURY/ PRIM |
|---|---|---|---|---|
| $RF$ | gC (kg soil)$^{-1}$ | FOM derived respiration when soil was incubated with FOM addition | yes | yes |
| $RS$ | gC (kg soil)$^{-1}$ | SOM derived respiration when soil was incubated with FOM addition | yes | yes |
| $RS_{Ctrl}$ | gC (kg soil)$^{-1}$ | SOM derived respiration when soil was incubated without FOM addition | yes | yes |
| Priming effect | gC (kg soil)$^{-1}$ | Differences between SOM derived respiration when soil was incubated with and without FOM addition | yes | yes |
| $B$ | gC (kg soil)$^{-1}$ | Total microbial biomass concentrations when soil was incubated with FOM addition | yes | no |
| $B_{Ctrl}$ | gC (kg soil)$^{-1}$ | Total microbial biomass concentrations when soil was incubated without FOM addition | yes | no |
| $B_{FOMr}$ | unitless | Proportions of FOM derived C in microbial biomass when soil was incubated with FOM addition | yes | no |

**Table 4. List of parameters with prescribed values.**

| Parameters | Units | Values | References |
|---|---|---|---|





| | | | |
|---|---|---|---|
| $Ea_{main}$ | kJ mol$^{-1}$ | 20 | van Iersel and Seymou, 2002 |
| $Ea_{KM}$ | kJ mol$^{-1}$ | 30 | Davidson and Janssens, 2006 |
| $Ea_{LM}$ | kJ mol$^{-1}$ | 37 | Wang et al., 2012 |
| $Ea_{LS}$ | kJ mol$^{-1}$ | 53 | Wang et al., 2012 |
| $Ea_{des}$ | kJ mol$^{-1}$ | 20 | Kaiser et al. 2001 |
| $Ea_{SA}$ | kJ mol$^{-1}$ | 42 | Assumed |
| $Ea_{SP}$ | kJ mol$^{-1}$ | 52 | Assumed |
| $Ea_{SS}$ | kJ mol$^{-1}$ | 47 | Allison et al., 2010 |
| $Ea_{ads}$ | kJ mol$^{-1}$ | 5 | Elshafei et al. 2009 |
| $Ea_{uptake}$ | kJ mol$^{-1}$ | 47 | Allison et al., 2010 |
| $Ke_{min}$ | unitless | 0.1 | Kaiser et al., 2014,2015 |

**Table 5. Number of parameters and observations used in optimization for each model.**

| Models | Number of parameters | Number of independent observations |
|---|---|---|
| ORCHIMIC | 22 | 75 |
| CENTURY | 4 | 60 |
| PRIM | 7 | 60 |

**Table 6. Descriptions of six ORCHIMIC variants with or without N dynamics and considering different combinations of MFTs.**

| ORCHIMIC variants | MFTs | C dynamics | N dynamics |
|---|---|---|---|
| C-MFT1 | One generalist | Yes | No |
| C-MFT2 | One FOM specialist and one SOM specialist | Yes | No |
| C-MFT3 | One generalist, one FOM specialist and one SOM specialist | Yes | No |
| CN-MFT1 | One generalist | Yes | Yes |
| CN-MFT2 | One FOM specialist and one SOM specialist | Yes | Yes |
| CN-MFT3 | One generalist, one FOM specialist and one SOM specialist | Yes | Yes |

**Table 7. Optimized values and uncertainties of parameters for the six variants of the ORCHIMIC model.**

| Parameters | Units | Prior values | C-MFT1 | C-MFT2 | C-MFT3 |
|---|---|---|---|---|---|
| Cost | | | 208 | 201 | 206 |
| $Adj_{SA}$ | unitless | 37 | 36±2 | 38±2 | 39±2 |
| $Adj_{SP}$ | unitless | 29 | 29±2 | 31±2 | 31±2 |




| Parameters | Units | Prior values | | | |
|---|---|---|---|---|---|
| $AvailCr$ | $10^{-3}$ | 4.1 | 5.3±0.8 | 4.2±0.8 | 4.2±0.8 |
| $b$ | unitless | 0.01 | 0.14±0.04 | 0.16±0.04 | 0.12±0.03 |
| $BAr$ | unitless | 0.3 | 0.26±0.08 | 0.41±0.09 | 0.36±0.09 |
| CAE | unitless | 0.6 | 0.81±0.09 | 0.79±0.08 | 0.85±0.09 |
| $d_{ENZ}$ | $10^{-3}$ h$^{-1}$ | 1 | 2.1±0.7 | 2.7±0.5 | 2.0±0.5 |
| $d_{MFT}$ | $10^{-3}$ h$^{-1}$ | 2 | 2.7±1.5 | 1.9±1.1 | 1.8±1.4 |
| $FEr$ | unitless | 0.1 | 0.45±0.19 | 0.50±0.19 | 0.45±0.18 |
| $K_{BA}$ | unitless | 6 | 6.2±1.9 | 8.1±2.0 | 8.7±2.0 |
| $K_{des}$ | $10^{-4}$ h$^{-1}$ | 10 | 28±10 | 6.7±2.9 | 9.6±3.5 |
| $Ke$ | $10^{-4}$ h$^{-1}$ | 0.1 | 1.3±0.9 | 2.3±1.2 | 0.93±0.54 |
| $Kr_{ref}$ | $10^{-3}$ h$^{-1}$ | 2 | 2.5±1.0 | 2.0±0.6 | 2.9±0.8 |
| $KM_F$ | gC (kg soil)$^{-1}$ | 50 | 77±20 | 50±19 | 57±19 |
| $KM_S$ | gC (kg soil)$^{-1}$ | 250 | 224±92 | 471±96 | 314±92 |
| $KM_{uptake}$ | gC (kg soil)$^{-1}$ | 0.26 | 13±4 | 13±5 | 15±5 |
| $LtoSS$ | unitless | 0.02 | 0.24±0.07 | 0.29±0.06 | 0.24±0.07 |
| $SEr$ | unitless | 0.1 | 0.46±0.17 | 0.70±0.18 | 0.61±0.17 |
| $Adsorb_{max}$ | gC (kg soil)$^{-1}$ | 1.35 | 3.1±0.7 | 2.1±0.7 | 3.2±0.8 |
| $Vmax_{LM}$ | g C (g ENZ C)$^{-1}$ h$^{-1}$ | 56 | 177±81 | 112±68 | 157±75 |
| $Vmax_{SS}$ | g C (g ENZ C)$^{-1}$ h$^{-1}$ | 1 | 7.5±5.7 | 13±7 | 18±9 |
| $Vmax_{uptake,C}$ | g C (g ENZ C)$^{-1}$ h$^{-1}$ | 0.24 | 0.74±0.27 | 0.29±0.13 | 0.52±0.20 |
| Parameters | Units | Prior values | CN-MFT1 | CN-MFT2 | CN-MFT3 |
| Cost | | | 203 | 201 | 218 |
| $Adj_{SA}$ | unitless | 37 | 37±2 | 38±2 | 37±2 |
| $Adj_{SP}$ | unitless | 29 | 32±2 | 31±2 | 29±2 |
| $AvailCr$ | $10^{-3}$ | 4.1 | 4.7±0.8 | 4.2±0.8 | 6.1±0.8 |
| $b$ | unitless | 0.01 | 0.18±0.05 | 0.16±0.04 | 0.24±0.08 |
| $BAr$ | unitless | 0.3 | 0.48±0.10 | 0.41±0.09 | 0.48±0.12 |
| CAE | unitless | 0.6 | 0.85±0.09 | 0.79±0.08 | 0.85±0.08 |
| $d_{ENZ}$ | $10^{-3}$ h$^{-1}$ | 1 | 1.8±0.5 | 2.7±0.5 | 1.5±0.8 |
| $d_{MFT}$ | $10^{-3}$ h$^{-1}$ | 2 | 2.3±1.2 | 1.9±1.1 | 2.6±1.2 |
| $FEr$ | unitless | 0.1 | 0.37±0.19 | 0.50±0.19 | 0.57±0.20 |
| $K_{BA}$ | unitless | 6 | 5.7±1.9 | 8.1±2.0 | 11±2 |
| $K_{des}$ | $10^{-4}$ h$^{-1}$ | 10 | 14±6 | 6.7±2.9 | 35±11 |





| | | | | | |
|---|---|---|---|---|---|
| $Ke$ | $10^{-4}$ h$^{-1}$ | 0.1 | 1.7±1.0 | 2.3±1.2 | 0.69±0.59 |
| $Kr_{ref}$ | $10^{-3}$ h$^{-1}$ | 2 | 2.0±0.6 | 2.0±0.6 | 1.6±0.6 |
| $KM_F$ | gC (kg soil)$^{-1}$ | 50 | 29±18 | 50±19 | 70±20 |
| $KM_S$ | gC (kg soil)$^{-1}$ | 250 | 401±96 | 471±96 | 120±93 |
| $KM_{uptake}$ | gC (kg soil)$^{-1}$ | 0.26 | 9.2±4.8 | 13±5 | 11±5 |
| $LtoSS$ | unitless | 0.02 | 0.27±0.07 | 0.29±0.06 | 0.14±0.08 |
| $SEr$ | unitless | 0.1 | 0.46±0.15 | 0.70±0.18 | 0.47±0.19 |
| $Adsorb_{max}$ | gC (kg soil)$^{-1}$ | 1.35 | 3.8±0.8 | 2.1±0.7 | 2.9±0.7 |
| $Vmax_{LM}$ | g C (g ENZ C)$^{-1}$ h$^{-1}$ | 56 | 83±65 | 112±68 | 190±86 |
| $Vmax_{SS}$ | g C (g ENZ C)$^{-1}$ h$^{-1}$ | 1 | 13±8 | 13±7 | 2.7±.3.3 |
| $Vmax_{uptake,C}$ | g C (g ENZ C)$^{-1}$ h$^{-1}$ | 0.24 | 0.29±0.17 | 0.29±0.13 | 0.48±0.22 |

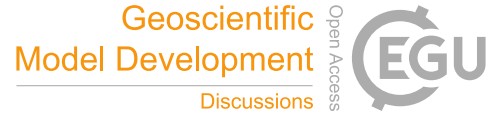

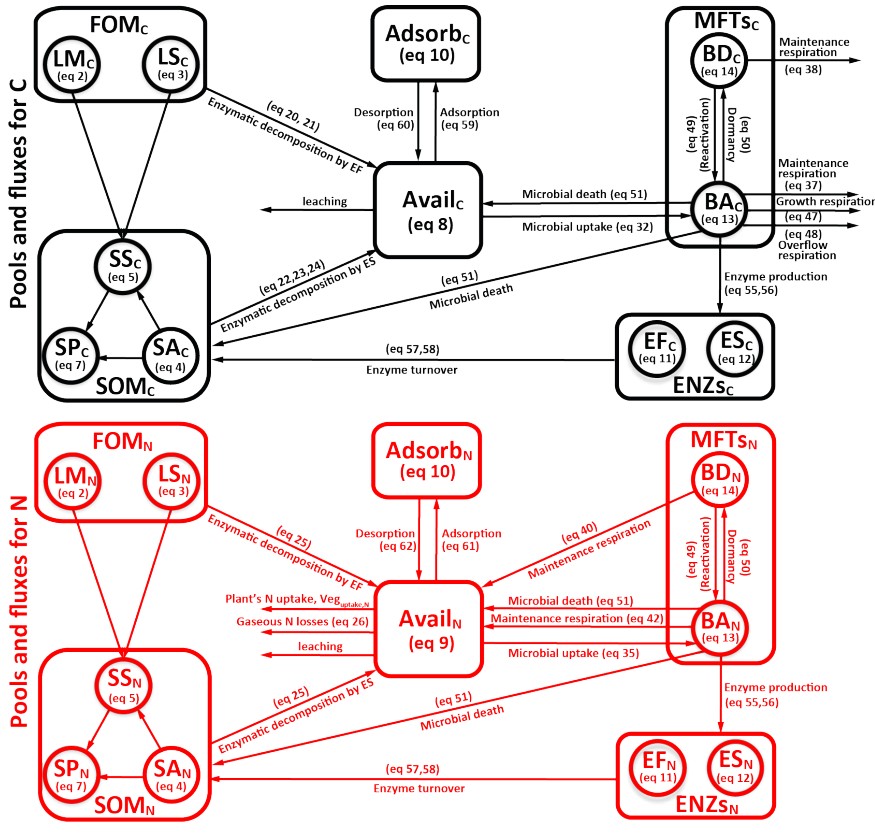

**FOM (Fresh organic matter pools):** LM: litter metabolic pool; LS: litter structural pool.

**SOM (soil oragnic matter pools):** SA: soil active pool; SS: soil slow pool; SP: soil passive pool.

**Avail:** pool includs available C and N to microbes.

**Adsorb:** pool includs available C and N adsorbed by mineral surface.

**ENZs (Enzyme pools):** EF: FOM decomposing enzyme; ES: SOM decomposing enzyme.

**MFTs (Microbial biomass pool):** BA: active microbial biomass; BD: dormant microbial biomass.

**Figure 1: Model structure of the ORCHIMIC. Rectangles and circles represent pools and arrows represent fluxes for C (black) and N (red). The carbon and nitrogen pools are described in Sect. 2.2. Equations describing the dynamics of each pool and flux are shown in brackets in the figure and can be found in Sect. 2.2 and 2.3. Arrows between FOM and SOM pools and within SOM pools represent fluxes due to physicochemical protection by mineral association and micro aggregate occlusion. $Veg_{uptake,N}$ is uptake of N by plants and is not explicitly simulated by ORCHIMIC.**



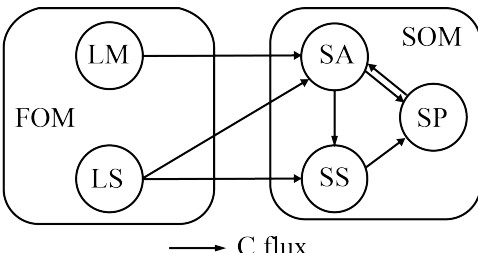

FOM: LM: litter metabolic pool; LS: litter structural pool;
SOM: SA: soil active pool; SS: soil slow pool; SP: soil passive pool;

**Figure 2: Pools and fluxes of the CENTURY and PRIM models.**

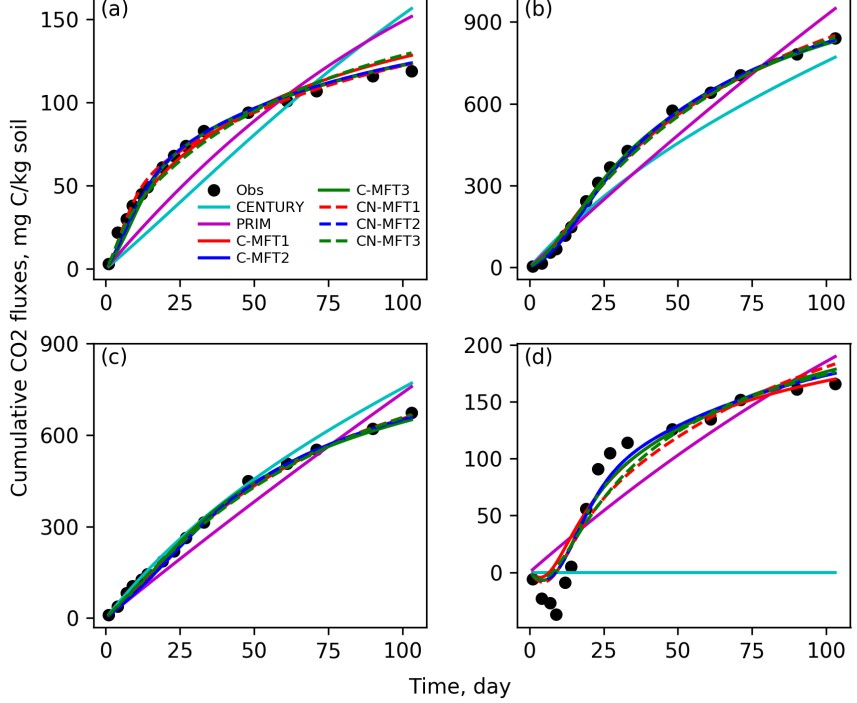

**Figure 3: Modelled and observed cumulative respiration from a) FOM, b) SOM with FOM addition c) SOM without FOM addition, and d) priming effect (difference between measured SOM derived respiration with FOM addition minus without FOM addition) (C-MFT2 overlapped with CN-MFT2).**





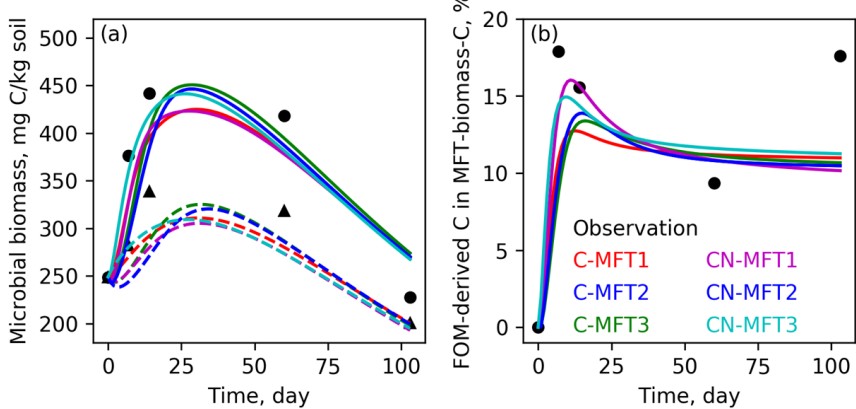

**Figure 4: Modelled and observed microbial biomass and proportion of FOM derived C in the biomass of different MFTs (curve for C-MFT2 overlapped with CN-MFT2). Solid lines show the evolution of microbial biomass or proportion of FOM derived C in MFT-biomass C with FOM addition and dashed lines show evolution of microbial biomass without FOM addition. Black filled circles and triangles are the observation with and without FOM addition, respectively.**

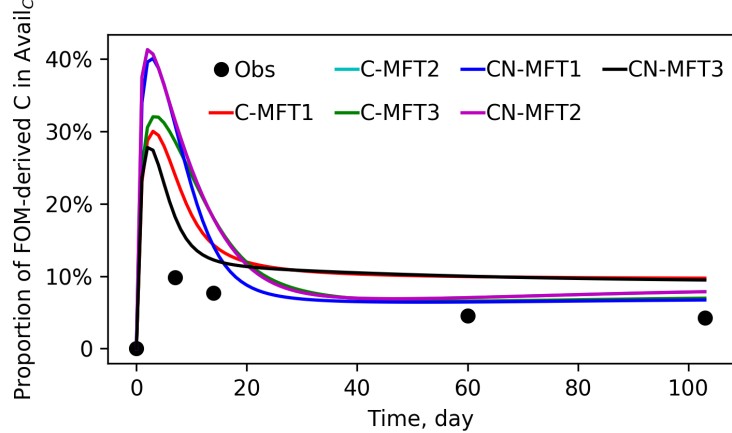

**Figure 5: Modelled and observed proportions of FOM derived C in the *Avail_C* pool. The curve of the C-MFT2 modelled curve overlaps with the one of CN-MFT2.**



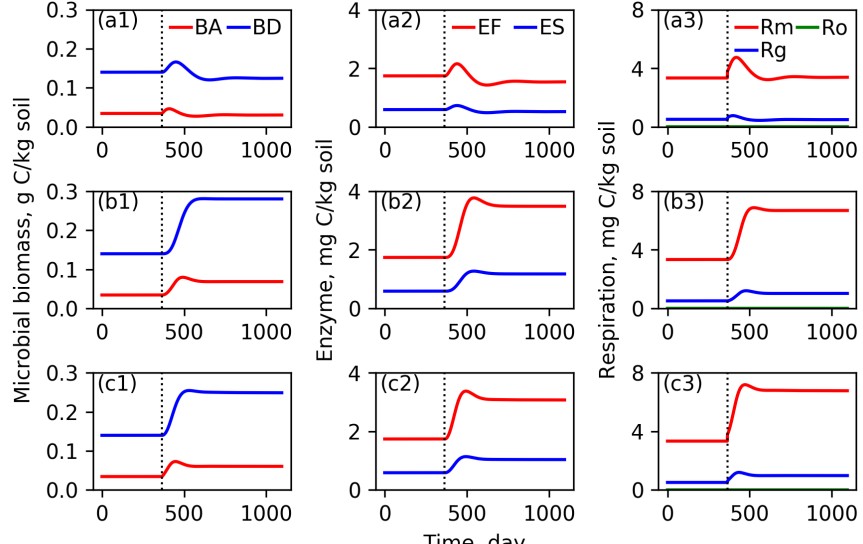

**Figure 6: Evolutions of active (*BA*) and dormant (*BD*) microbial biomass, FOM decomposing enzymes (*EF*) and SOM decomposing enzymes (*ES*), maintenance respiration (*Rm*), growth respiration (*Rg*) and overflow respiration (*Ro*) for CN-MFT3 (standard version of ORCHIMIC) when temperature is stepwise increased by 5 K (a1, a2 and a3), when FOM input doubles (b1, b2 and b3), and when both forcings are changed (c1, c2 and c3). The vertical black dotted line shows the time when the stepwise increase of temperature and/or the doubling FOM input was implemented.**





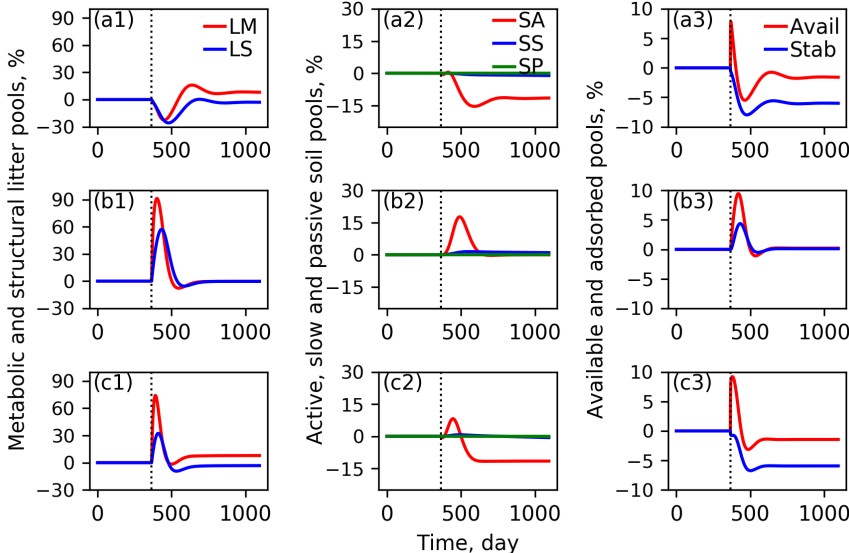

**Figure 7: Relative changes of C in metabolic (*LM*) and structural (*LS*) litter pools (a1, b1 and c1), in active (*SA*), slow (*SS*) and passive (*SP*) soil pools (a2, b2 and c2), and in available (*Avail*) and absorbed (*Absorb*) pools (a3, b3 and c3) for the CN-MFT3 model when temperature is stepwise increased by 5 K (a1, a2 and a3), when FOM input doubles (b1, b2 andb3), and both (c1, c2 and c3). The vertical black dotted line shows the time when the change of temperature and/or FOM input was implemented.**





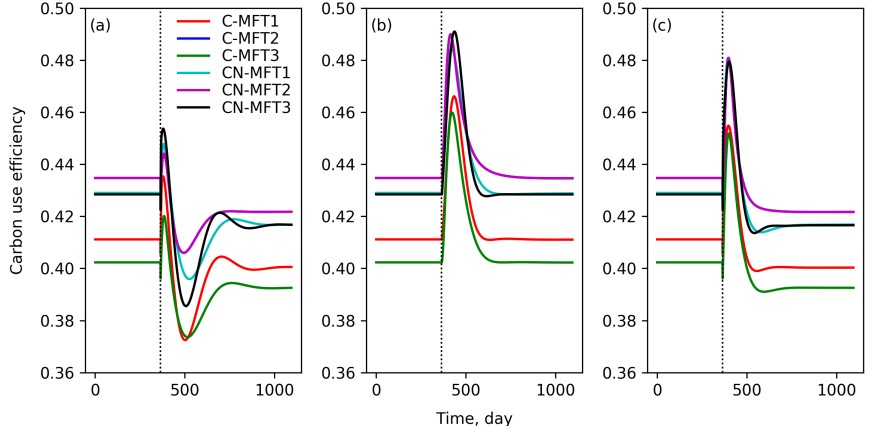

**Figure 8. Temporal evolution of carbon use efficiencies (defined as ratio of carbon allocated to microbial growth to the sum of those allocated to growth and respiration) when temperature is stepwise increased by 5 K (a), when FOM input doubles (b), and when both forcings are changed (c), for the six variants of the ORCHIMIC model. The curve for C-MFT2 overlapped with CN-MFT2. The vertical black dotted line shows the time when the change of temperature and/or input was applied.**