# Peer review of "ORCHIMIC (v1.0), a microbe-driven model for soil organic matter decomposition designed for large-scale applications"

_Geoscientific Model Development, 2017_

## Referee Comment (RC1) · Anonymous Referee #1 · 11 Mar 2018

The authors integrated the state-of-the-art knowledge into a microbe-enabled soil C-N model (ORCHIMIC) with four microbial functional groups and a dynamic enzyme production approach. I acknowledge the authors for a comprehensive comparison of their results to literature and observations in terms of parameter values and pool sizes. To my understanding, the "dynamic enzyme production mechanism" is a vital improvement over existing models. Thus I would like the authors, if possible, to address this feature in a separate sub-section in "Discussion". My second suggestion is to show the simulated C:N ratios in major organic matter and microbial pools (you may include them as supplementary information) as ORCHIMIC is a C-N coupled model. One more suggestion for this paper is to use simple notations in most of the equations. For

example, you may use the notation (F(T)) for the temperature functions in Eqs. (20-24) and add one more equation for F(T) as you did for Eq. (15-19). In addition, it is negotiable whether it is appropriate to specifically address "large-scale applications" in the title, as it was only tested with lab-incubation data in this paper, although it is known that ORCHIMIC may be coupled to large-scale models. Other minor comments: Abstract: Page 1, Line 32: I think "the model" refers to the ORCHIMIC model, please explicitly use the model name.

---

## Referee Comment (RC2) · Anonymous Referee #2 · 15 Mar 2018

The authors provide a well-documented new ORCHIDEE-family model that introduces a number of features: explicit microbial biomass pool, dormancy, MFTs, coupled C and N dynamics, and mineral protection. The model reasonably reproduces $CO_2$ fluxes and microbial biomass measurements. I have some comments and questions below that I hope are helpful.

p.4 L4 says that this model is embedded in the land model ORCHIDEE but it is also a zero-D model - wouldn't embedding in a land model make it at least 2-D? If you are not using the land model feature for this study, I would hesitate to say this.

p.4, L25 mentions fluxes that represent occlusion by macro-aggregates but it is not

clear from the model description which fluxes these would be (since there is not specifically an aggregate pool) in the way that, for example, adsorption is clearly labeled in the conceptual diagram. I would either be clear that aggregates are implicitly represented by the exchanges between the SOM pools or clearly define which exchange is meant.

p.5, L27 I'm fine with the implicit representation of cheaters, but I am curious how much coexistence you achieved between MFTs in the multiple MFT models. The same limitations that you describe with cheaters can apply, causing all but one MFT to quickly die off in spatially or temporally homogenous environments, e.g., zero-D simulation or constant environmental forcing, respectively.

p.9, L25 There is generally good attribution of where functional forms and parameters for equations come from, but there are a few places where it is unclear. For example, is Equation 15 taken from Parton et al. 1987 or somewhere else? Some of these very empirical forms need to be either cited or explained. Further into this, the water-modifying equations for decomposition (Eq 15) and uptake (Eq 28) look very different from one another. Why is that?

p.12, L2 I think there is a way to avoid performing the adjustment in Eq 32. It involves including total available C as a term in your uptake rate calculation (Eq. 31), but in order to get the multiple MFT competition to scale correctly, you would need something like [Tang and Riley, 2017]. I'm not recommending this for this paper, but something to think about for the future.

p.21, L29 No change in SOM after doubled inputs is a common observation with microbial models [Wang et al., 2013, 2015] because your microbial death rate (Eq 51) is linear. If it were density-dependent (i.e. BAd = dMFT*BA^2*dt), then you would likely see some response to increased inputs because microbial biomass would no longer be exactly proportional to inputs (see [Georgiou et al., 2017]). Not necessary to change your model, as many models use linear turnover, but I think it is important to acknowledge the potential importance of this choice to the model behavior somewhere in the text.

References:

Georgiou, K., R. Z. Abramoff, J. Harte, W. J. Riley, and M. S. Torn (2017), Microbial community-level regulation explains soil carbon responses to long-term litter manipulations, Nat. Commun., 8(1223), 1–10, doi:10.1038/s41467-017-01116-z.

Tang, J. Y., and W. J. Riley (2017), SUPECA kinetics for scaling redox reactions in networks of mixed substrates and consumers and an example application to aerobic soil respiration, Geosci. Model Dev., 10(9), 3277–3295, doi:10.5194/gmd-10-3277-2017.

Wang, G., W. M. Post, and M. a. Mayes (2013), Development of microbial-enzyme-mediated decomposition model parameters through steady-state and dynamic analyses, Ecol. Appl., 23(1), 255–272, doi:10.1890/12-0681.1.

Wang, Y. P. et al. (2015), Responses of two nonlinear microbial models to warming or increased carbon input, Biogeosciences Discuss., 12(17), 14647–14692, doi:10.5194/bgd-12-14647-2015.
* * *

---

## Author Comment (AC1) · 7 May 2018

**Responses to comments from Referee 1**

General comment

The authors integrated the state-of-the-art knowledge into a microbe-enabled soil C-N model (ORCHIMIC) with four microbial functional groups and a dynamic enzyme production approach. I acknowledge the authors for a comprehensive comparison of their results to literature and observations in terms of parameter values and pool sizes.

Response

Thanks for your valuable comments. We revised the manuscript accordingly. Please see the following responses to all the comments.

Comment 1

To my understanding, the "dynamic enzyme production mechanism" is a vital improvement over existing models. Thus I would like the authors, if possible, to address this feature in a separate sub-section in "Discussion".

Response

Following sub-section was added to the discussion section:

"7.4 Dynamic enzyme production

Unlike in some microbial models where enzyme production depends solely on microbial biomass or microbial uptake, the saturation level of substrate is an important factor affecting enzyme production in ORCHIMIC. Microbes increase enzyme production if there is more substrate available to grow faster and decrease enzyme production when substrate is depleting to avoid unnecessary allocation of C and N to the enzyme production function. In ORCHIMIC, the saturation level of directly available C also affects enzyme production. Enzyme production per unit of microbial biomass decreases with increasing available C (see Eq. (53)), e.g., by catabolic repression of enzyme synthesis by the product of the reaction. This also corresponds to the fact that the fraction of cheaters - microbes that do not produce enzymes – increases with increasing available C. Cheaters were added as an explicit microbial functional group in an individual-based micro-scale microbial community model with an explicit positioning of microbes to access substrate (Kaiser el al., 2015). Such an approach is only applicable in a micro scale model, because the coexistence of cheaters and enzyme-producing microbes is only sustainable in heterogeneous environments. In non-spatially explicit zero-dimensional models, like ORCHIMIC that assumes a homogeneous environment, cheaters will always have a competitive advantage over other microbes in taking up

C and N while not having to invest in enzyme production. This will eventually drive enzyme-producing MFTs to extinction at steady state (Allison, 2005), the model will not produce enzymes anymore and all microbes will die in the end. With the dynamic enzyme production mechanism described in equations Eq. (51)-(55), cheaters can be included in ORCHIMIC with a possible co-existence with non-cheaters microbes in the model, though cheaters are not parameterized in an explicitly way as a separate MFT group."

Comment 2

My second suggestion is to show the simulated C:N ratios in major organic matter and microbial pools (you may include them as supplementary information) as ORCHIMIC is a C-N coupled model.

Response

Three figures (Fig. S13-15) showing C/N ratios for microbial, soil organic matter, available and absorbed pools in the idealized simulations with increasing FOM input and/or increasing temperature for the three ORCHIMIC variants with N dynamics were added in the supporting information. It should be noted that, because in the idealized simulations the rate and C/N ratio of litter input are prescribed, and decomposition does not feedback to the C/N ratio of the substrate, the C/N ratios for the two litter pools are constant during the simulations and were not shown.

Comment 3

One more suggestion for this paper is to use simple notations in most of the equations. For example, you may use the notation (F(T)) for the temperature functions in Eqs. (20-24) and add one more equation for F(T) as you did for Eq. (15-19).

Response

The temperature functions in Eq. 20-24 and also other equations with temperature sensitivity function used for parameter KM were all simplified as $F_{T,KM}$. The expression

$F_{T,KM}$ is described in Eq. 16 in the revised ms.

Comment 4

In addition, it is negotiable whether it is appropriate to specifically address "large-scale applications" in the title, as it was only tested with lab-incubation data in this paper, although it is known that ORCHIMIC may be coupled to large-scale models.

Response

The title was revised as "ORCHIMIC (v1.0), a microbe-mediated model for soil organic matter decomposition".

Comment 5

Other minor comments: Abstract: Page 1, Line 32: I think "the model" refers to the ORCHIMIC model, please explicitly use the model name.

Response

Revised accordingly.

**Responses to comments from Referee 2**

General comment

The authors provide a well-documented new ORCHIDEE-family model that introduces a number of features: explicit microbial biomass pool, dormancy, MFTs, coupled C and N dynamics, and mineral protection. The model reasonably reproduces CO2 fluxes and microbial biomass measurements. I have some comments and questions below that I hope are helpful.

Response

Thank you for reviewing this manuscript. We revised the manuscript accordingly. Please see the following responses to the comments.

[Figure]

Comment 1

p.4 L4 says that this model is embedded in the land model ORCHIDEE but it is also a zero-D model - wouldn't embedding in a land model make it at least 2-D? If you are not using the land model feature for this study, I would hesitate to say this.

Response

Currently, ORCHIMIC v0 is a zero-D model and is independent from ORCHIDEE, but its input data is consistent with the output of ORCHIDEE. The last sentence in the second last paragraph has been revised as "ORCHIMIC is developed with the aim of being incorporated in the ORCHIDEE land biosphere model (Krinner et al., 2005), but its generic input data would allow it to be embedded in any other global land surface models for grid-based simulations."

Comment 2

p.4, L25 mentions fluxes that represent occlusion by macro-aggregates but it is not clear from the model description which fluxes these would be (since there is not specifically an aggregate pool) in the way that, for example, adsorption is clearly labeled in the conceptual diagram. I would either be clear that aggregates are implicitly represented by the exchanges between the SOM pools or clearly define which exchange is meant.

Response

The fluxes representing macro-aggregates are implicitly modeled in ORCHIMIC. The corresponding sentence has been revised as "Besides, there are fluxes from FOM pools to SS pool, from SA to both SS and SP pools and from SS to SP pool to implicitly represent physicochemical protection mechanisms, such as occlusion of substrates in macro-aggregates (Parton et al., 1987)."

Comment 3

p.5, L27 I'm fine with the implicit representation of cheaters, but I am curious how much coexistence you achieved between MFTs in the multiple MFT models. The same limitations that you describe with cheaters can apply, causing all but one MFT to quickly die off in spatially or temporally homogenous environments, e.g., zero-D simulation or constant environmental forcing, respectively.

Response

Yes, you are right. The same limitation can also apply for other MFTs. In our multiple MFT model, the microbial pool could end up with one dominant MFT depending on which substrate- FOM or SOM – is the major C and N source. In our idealized simulation, at equilibrium, the dominant MFT can reach 99% of total microbial pool because of the constant environmental forcing applied. However, the major purpose of having several MFTs is not to model the coexistence of MFTs, but to make it possible to have different combination of MFTs, thus different C/N ratio of microbial pool, under different environments and variable environmental conditions. In Sect. 7.3, the last sentence has been revised as "Also, different major sources (FOM or SOM) of C and N favor different MFTs depending on their enzyme production cost. Thus, with more than one MFT, the C/N ratio of the microbial pool can be variable (Fig. S13-15)."

Comment 4

p.9, L25 There is generally good attribution of where functional forms and parameters for equations come from, but there are a few places where it is unclear. For example, is Equation 15 taken from Parton et al. 1987 or somewhere else? Some of these very empirical forms need to be either cited or explained.

Response

The Equation 15 is taken from ORCHIDEE (Krinner et al., 2005) and the references have been added for Eq. 15-19.

Ref.: Krinner, G., Viovy, N., de Noblet-Ducoudré, N., Ogée, J., Polcher, J., Friedlingstein, P., Ciais, P., Sitch, S., Prentice, I.C.: A dynamic global vegetation model for studies of the coupled atmosphere-biosphere system, Global Biogeochem. Cycles, 19, GB1015, 2005

Comment 5

Further into this, the water- modifying equations for decomposition (Eq 15) and uptake (Eq 28) look very different from one another. Why is that?

Response

The water-modifying equations for uptake (Eq. 28) is actually not used in ORCHIMIC model as the effect of water has already been considered during decomposition. Eq. 28 has been tested at some points and was integrated in a previous version of the manuscript and has not been deleted. We apologize for this error and Eq. 28 has been removed from the text.

Comment 6

p.12, L2 I think there is a way to avoid performing the adjustment in Eq 32. It involves including total available C as a term in your uptake rate calculation (Eq. 31)

Response

Thank you for the suggestion. However, if we include the total available C as a term in the uptake rate calculation, the parameter will be time step dependent, because the C becoming available from decomposition is time step dependent. This may lead to instabilities quite complex to fix. We tried to avoid this.

Comment 7

but in order to get the multiple MFT competition to scale correctly, you would need something like [Tang and Riley, 2017]. I'm not recommending this for this paper, but something to think about for the future.

Response

Thanks for this nice suggestion, and we will consider it in the future development of ORCHIMIC model.

Comment 8

p.21, L29 No change in SOM after doubled inputs is a common observation with microbial models [Wang et al., 2013, 2015] because your microbial death rate (Eq 51) is linear. If it were density-dependent (i.e. BAd = dMFT*BAЁĘ2*dt), then you would likely see some response to increased inputs because microbial biomass would no longer be exactly proportional to inputs (see [Georgiou et al., 2017]). Not necessary to change your model, as many models use linear turnover, but I think it is important to acknowledge the potential importance of this choice to the model behavior somewhere in the text. References: Georgiou, K., R. Z. Abramoff, J. Harte, W. J. Riley, and M. S. Torn (2017), Microbial community-level regulation explains soil carbon responses to long-term litter manipulations, Nat. Commun., 8(1223), 1–10, doi:10.1038/s41467-017-01116-z. Tang, J. Y., and W. J. Riley (2017), SUPECA kinetics for scaling redox reactions in networks of mixed substrates and consumers and an example application to aerobic soil respiration, Geosci. Model Dev., 10(9), 3277–3295, doi:10.5194/gmd-10-3277- 2017. Wang, G., W. M. Post, and M. a. Mayes (2013), Development of microbial-enzyme- mediated decomposition model parameters through steady-state and dynamic analyses, Ecol. Appl., 23(1), 255–272, doi:10.1890/12-0681.1. Wang, Y. P. et al. (2015), Responses of two nonlinear microbial models to warming or increased carbon input, Biogeosciences Discuss., 12(17), 14647–14692, doi:10.5194/bgd-12-14647-2015.

Response

Thanks for bringing up this point. In Sect. 7.2, the third last sentence in the last paragraph was revised as: "These responses were different from the proportional increase in soil C pools as modelled by the conventional linear SOC decomposition

model, but consistent with those observed from microbial models with linear microbial death rate (Wang et al., 2013; Wang et al., 2016). It should be noted that with density-dependent microbial mortality, the growth of microbes with an increase of FOM input might be limited and lead to accumulation of soil C (Georgiou et al., 2017)."

Please also note the supplement to this comment:
https://www.geosci-model-dev-discuss.net/gmd-2017-325/gmd-2017-325-AC1-supplement.zip